# Many plants naturalized as aliens abroad have also become more common within their native regions

Rashmi Paudel [1,2] ✉, Trevor S. Fristoe[1,3], Nicole L. Kinlock[1], Amy J. S. Davis [1], Weihan Zhao [1,2], Hans Van Calster [4], Milan Chytrý [5], Jiří Danihelka [5,6], Guillaume Decocq[7], Luise Ehrendorfer - Schratt[8], Kun Guo [9], Wen-Yong Guo [10,11,12], Zdeněk Kaplan [6,13], Simon Pierce[14], Jan Wild[15], Wayne Dawson [16], Franz Essl[8,17], Holger Kreft [18,19,20], Jan Pergl [21], Petr Pyšek [21,22], Marten Winter [23] & Mark van Kleunen [1,24]

Due to anthropogenic pressure some species have declined whereas others have increased within their native ranges. Simultaneously, many species introduced by humans have established self-sustaining populations elsewhere (i.e. have become naturalized aliens). Previous studies have shown that particularly plant species that are common within their native range have become naturalized elsewhere. However, how changes in native distributions correlate with naturalization elsewhere is unknown. We compare data on grid-cell occupancy of native vascular plant species over time for 10 European regions (countries or parts thereof). For nine regions, both early occupancy and occupancy change correlate positively with global naturalization success (quantified as naturalization in any administrative region and as the number of such regions). In other words, many plant species spreading globally as naturalized aliens are also expanding within their native regions. This implies that integrating data on native occupancy dynamics in invasion risk assessments might help prevent new invasions.

Natural processes such as tectonic upheavals and glaciations or evolutionary innovations have driven dynamics in species distributions throughout the history of life[1,2]. In the last centuries, however, and particularly since the start of the Anthropocene in the mid-20th century[3], human pressures, such as land-use change, habitat fragmentation and eutrophication, have accelerated these dynamics dramatically[4]. In particular, human pressures have caused rapid declines in populations and range sizes of many native species[5], resulting in drastic declines in occupancy (i.e., the number of locations of occurrence) in many regions where those species are native e.g.[6–8]. However, while many species are in decline, and c. 25% of assessed animal and plant species are threatened with extinction[4], there are also species that benefit from the rapid changes occurring during the Anthropocene and are on the rise[9].

Data on temporal changes in regional occupancy for large numbers of species are still relatively rare and restricted to European regions. Nevertheless, the few studies that have analyzed such data consistently show that native species with increasing occupancy are typically tall, habitat generalists, classified as competitors in Grime's CSR-strategy framework, and with high values of the Ellenberg indicator for nitrogen[6,10–12]. This corroborates the idea that these species have benefited from anthropogenic environmental changes, such as atmospheric nitrogen deposition[6] and the creation of novel anthropogenic habitats[13] within their native ranges.

Concurrent with anthropogenic changes to the environment, humans have intentionally and unintentionally transported many species from their native ranges across major geographical barriers to new regions[14]. Though most of these introduced alien species have

failed to establish self-sustaining populations outside captivity or cultivation, a small percentage have succeeded in becoming naturalized, occasionally in hundreds of regions (e.g., countries, states and provinces) around the globe[15–17]. Among the vascular plants, more than 16,000 species have already become naturalized somewhere on Earth[18], provisionally accepted]. Most of these naturalizations happened after the 1950s, i.e., during the Anthropocene[19], and predominantly in habitats with high levels of anthropogenic disturbances e.g.[20]. Previous analyses of the characteristics of naturalized plants or the subset of invasive alien plants (i.e., naturalized plants that have spread rapidly and frequently have ecological and/or socio-economic impacts) have shown that they are typically tall, habitat generalists, classified as competitors in Grime's CSR-strategy framework, and with high values of the Ellenberg indicator for nitrogen e.g.[12,21–25]. – just like the species that are increasing within their native ranges[26,27]. However, whether species that have increased their occupancy in regions of their native range and species that have become naturalized elsewhere are largely the same species has never been tested explicitly. If this is the case, it would imply that information on native occupancy dynamics could inform invasion risk assessments.

Ultimately, occupancy dynamics of species, both within and outside their native range, are likely to depend on intrinsic features of the species. Unfortunately, despite the many studies that have measured plant functional traits, for most traits, data are available only for a small proportion of the global flora[28]. A notable exception is woodiness, which is indicative of both growth form (i.e., woody species are usually shrubs or trees) and habitat affiliation (i.e., woody species typically occur in forests and other closed habitats). On the one hand, the tall stature of woody species allows them to have greater dispersal capacities and a higher competitive dominance, which could facilitate range expansion in both native and non-native regions. However, despite these potential advantages, woody species generally have a lower probability of naturalization than non-woody species[29]. This may be because woody plants usually are less successful in frequently disturbed habitats, and because they have longer generation times and therefore require more time to become naturalized and to spread after introduction[30,31]. So, species features, such as woodiness, might mediate the relationship between global naturalization success and occupancy dynamics in the native range.

It has been suggested that extinction risk of native species and invasion success of alien species might represent two sides of the same coin[32]. Jeschke and Strayer[33] did not find this to be the case for birds and freshwater fish. However, this concept not yet been assessed in plants. The findings that naturalized plants and those spreading in their native range share a common set of traits suggests that it may be the case[6,21–27]. Accordingly, numerous studies have shown that

common species with large native ranges are more likely to naturalize elsewhere[34,35]. However, range size is just one dimension of commonness. Other dimensions include habitat breadth and local abundance, as proposed by Rabinowitz[36], and occupancy, as recently proposed by Crisfield et al.[37]. These different dimensions of commonness are frequently positively correlated[38,39], and some studies have shown that regional occupancy in the native range correlates positively with naturalization success elsewhere[21,22,25]. However, in addition to these static measures of commonness, the change in occupancy over time could be considered a further dimension, similar to spread rate, which has been proposed as one of the dimensions of invasiveness for alien species[40]. If species that increase their occupancy within their native range over time are largely the same species that spread as naturalized aliens elsewhere, this would suggest that similar mechanisms may underlie both processes.

Here, we test the hypothesis that the plant species that have become widely naturalized across the globe are also increasing in occupancy (i.e., in the proportion of grid cells in which they have been recorded) within their native regions. To test this hypothesis, one would ideally have time series data of grid-cell occupancies for the entire native range of the species. However, as such data is not available, we instead retrieve data on grid-cell occupancies of vascular plant species during an early period (i.e., early occupancy) and a later period, each usually covering multiple decades, for 10 regions in Europe (Fig. 1). For each of these 10 native regions, which correspond to countries or parts thereof, we calculate for each native species an occupancy-change index according to Telfer et al.[41]. This index quantifies the degree to which the change in the proportion of grid cells in which a species has been recorded between the early and later period is higher or lower than expected based on the early occupancy[41]. Consequently, the occupancy-change index is not correlated with early occupancy. We then use hurdle models to analyze how global naturalization success – a combination of naturalization incidence (i.e. whether or not a species has become naturalized, which is modeled using a Bernoulli distribution) and naturalization extent (i.e. the number of regions where a naturalized species has become naturalized, which is modeled using a zero-truncated negative binomial distribution) – correlates with occupancy in the early period and the occupancy change within the species' native regions. Given that the available data for the 10 native regions vary in many aspects (Table 1), the analyses are done for each region separately.

## Results
As we had data on woodiness of all 3920 species in our 10 datasets (see Supplementary Methods for details), we ran hurdle models with and without woodiness of the species as an additional predictor. However, as the results for our two main predictors of interest, occupancy in the early period and occupancy change within the species' native regions, remained largely the same, we focus here on the models without woodiness. Results for the models with woodiness are provided in Table S1 (also see Tables S2–S11).

### Naturalization success vs early occupancy in native range
Our hurdle models showed that global naturalization success was associated with occupancy in the early period for all 10 native regions (Table 2 (also see Tables S12–S21), Fig. 2). This was true for both the likelihood of being naturalized outside the native range (i.e. for the Bernoulli part of the hurdle model) and for being naturalized in more regions (i.e., for the zero-truncated count part of the hurdle model; Table 2, Fig. S1, S2).

### Naturalization success vs occupancy changes in native range
Species with high occupancy-change indices also had a higher likelihood of being naturalized for seven of the 10 native regions and were naturalized in more regions globally for nine of the 10 native regions

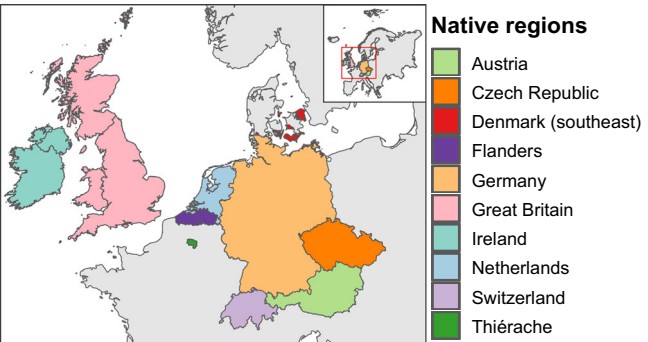

**Native regions**
- Austria
- Czech Republic
- Denmark (southeast)
- Flanders
- Germany
- Great Britain
- Ireland
- Netherlands
- Switzerland
- Thiérache

**Fig. 1 | Map showing the 10 European native regions.** For these regions, we have information on occupancies of native plant species for an early and a later time period. Polygons were obtained from GADM, the GloNAF database, or were created using Google Earth Pro (Data SIO, NOAA, U.S. Navy, NGA, GEBCO; Image Landsat / Copernicus).

## Table 1 | Details on the datasets for each region used in this study

| Region | Data source | Early period | Later period | Total number of grid cells | Size of grid cell | Number of native species used in analysis |
|---|---|---|---|---|---|---|
| Austria | Floristische Kartierung Österreichs[74] | before 1990 | 1990–2022 | 2600 | 5′ × 3′ (c. 6.25 × 5.55 km²) | 2419 |
| Czech Republic | Pladias Database of the Czech Flora and Vegetation (http://www.pladias.cz)[75] | before 2000 | 2001–2022 | 2551 | 5′ × 3′ (c. 6.25 × 5.55 km²) | 1834 |
| Denmark (southeastern regions) | [7] | 1857–1883 | 2015 | 263 | 5 × 5 km² | 921 |
| Flanders (northern Belgium) | [76] | 1939–1971 | 1972–2004 | 985 | 4 × 4 km² | 861 |
| Germany | 8 | 1960–1987 | 1997–2017 | 12024 | 5′ × 3′ (c. 6.25 × 5 km²) | 1715 |
| Great Britain | Plant Atlas 2020[77] | 1970–1986 | 2000–2019 | 2852 | 10 × 10 km² | 1355 |
| Ireland | Plant Atlas 2020[77] | 1987–1999 | 2000–2019 | 1007 | 10 × 10 km² | 910 |
| The Netherlands | Nationale Databank Flora en Fauna[78] | before1990 | 1990–2022 | 1685 | 5 × 5 km² | 1115 |
| Switzerland | Infoflora (The National Data and Information Center on the Swiss Flora; https://www.infoflora.ch/)[79] | before 2000 | 2000–2022 | 1827 | 5 × 5 km² | 2307 |
| Thiérache (northern France) | 10 | 1880–1900 | 1957–2005 | 129 | 4 × 4 km² | 775 |

## Table 2 | Overview of the estimates of the two-part hurdle models for the 10 native regions

| | Austria | Czech Republic | Denmark (southeast) | Flanders (Belgium) | Germany | Great Britain | Ireland | Netherlands | Switzerland | Thiérache (northern France) |
|---|---|---|---|---|---|---|---|---|---|---|
| *Bernoulli part* | | | | | | | | | | |
| Early occupancy (EO) | 0.76*** | 1.16*** | 0.86*** | 1.59*** | 1.35*** | 1.28*** | 1.02*** | 1.12*** | 0.75*** | 1.17*** |
| Occupancy change (OC) | 0.15** | 0.21** | 0.67*** | 0.60*** | 0.16* | 0.11 | 0.04 | 0.72*** | 0.27*** | 0.23 |
| EO × OC | −0.09 | 0.40*** | −0.007 | 0.17 | 0.14 | 0.06 | −0.08 | 0.40* | 0.45*** | 0.22 |
| *Zero-truncated count part* | | | | | | | | | | |
| Early occupancy (EO) | 0.36*** | 0.57*** | 0.49*** | 0.53*** | 0.66*** | 0.53*** | 0.53*** | 0.53*** | 0.38*** | 0.35*** |
| Occupancy change (OC) | 0.15*** | 0.09* | 0.27*** | 0.26*** | 0.17*** | 0.23*** | 0.18* | 0.45*** | 0.24*** | 0.02 |
| EO × OC | −0.05 | 0.08 | 0.04 | −0.03 | −0.06 | 0.01 | −0.003 | 0.21** | 0.27*** | 0.02 |

The Bernoulli part considers whether a species has become naturalized, and the zero-truncated count part considers the number of regions where a naturalized species has become naturalized. These models relate the two components of global naturalization success (naturalization incidence and extent) to occupancy in the early period (EO), the occupancy-change index (OC) and their interaction. To increase comparability, early occupancy was scaled to mean of zero and standard deviation of one. ***: $P < 0.001$, **: $P < 0.01$, *: $P < 0.05$. All $p$-values are two-sided, and no adjustments for multiple comparisons were applied. The detailed results of the hurdle models with the exact $p$-values for each region are provided in Tables S12–S21. Source data are provided as a Source Data file.

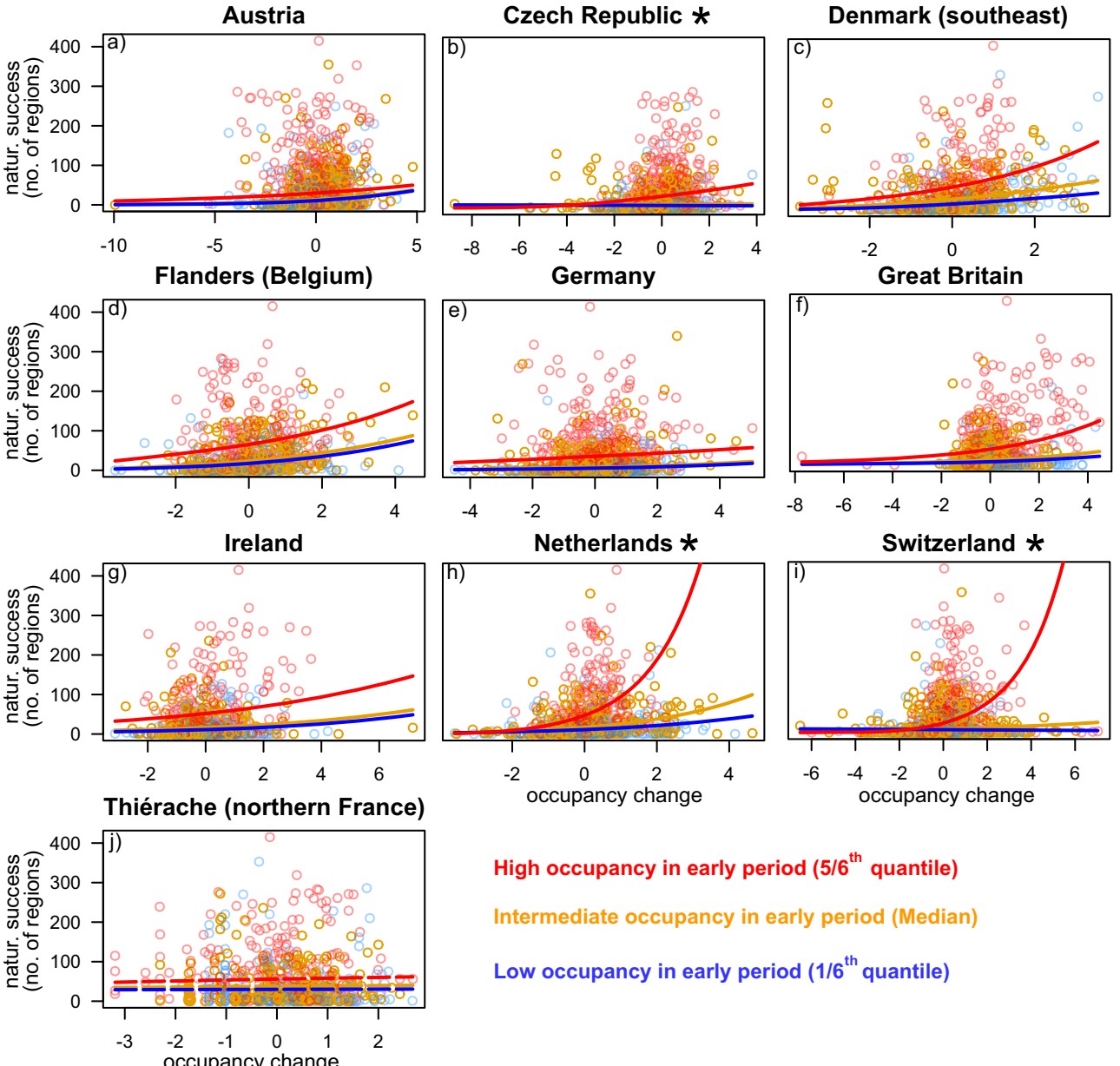

**Fig. 2 | Predicted relationships between global naturalization success and the occupancy-change index from hurdle models for the 10 native regions (a–j).** These models combined naturalization incidence (i.e., whether or not a species has become naturalized; Bernoulli distribution) and naturalization extent (i.e., the number of non-native regions where the species has become naturalized; zero-truncated negative binomial distribution). To illustrate how global naturalization success depends on early occupancy in the native region, the data points are colored according to whether they are in the upper, middle or lower third of the early occupancy distribution. Accordingly, the predicted relationships are plotted for early occupancy values set equal to the 5/6th quantile, the median and the 1/6th quantile. Significant relationships between global naturalization success and occupancy change (either for the Bernoulli or zero-truncated count part) are plotted with solid lines, and non-significant relationships are plotted in dashed lines. The regions for which the interaction between early occupancy and occupancy change was significant are marked with an asterisk (*) next to the region names. Source data are provided as a Source Data file.

(Table 2, Fig. 2, Fig. S1, S2). The Thiérache region was the only region where the occupancy-change index was not significantly associated with any of the two components of global naturalization success (Table 2, Fig. 2j, Fig. S1j, S2j). So, overall, our findings indicate that species that have increased in occupancy in their native regions relative to other species with similar early occupancy are also more likely to have become widely naturalized. For three of the 10 native regions (the Czech Republic, the Netherlands and Switzerland), the positive association between global naturalization success (either measured as the likelihood of being naturalized or as being naturalized in more regions) and the occupancy-change index was strongest for species that already had high occupancies in the early period (significant early

occupancy × occupancy change interactions in Table 2, Fig. 2b, h, i, Fig. S1b, h, i, S2h, i). These results thus indicate that recent increases in occupancy within these three native regions are particularly strongly associated with global naturalization success for species that were already common decades ago.

## Discussion

Our analyses show that global naturalization success is highest not only for species that already had high occupancies in their native European regions decades to centuries ago —when many of them were first introduced to new regions[19] — but also for species that have since then increased in occupancy within their native regions (Table 2).

Furthermore, for three of the 10 native regions, the positive association between global naturalization success and the occupancy-change index was particularly pronounced for species that had already a high occupancy in the native region during the early time period (Table 2, Fig. 2b, h, i). Our results thus show that many species that have increased their occupancy within their native regions have also increased their ranges abroad, whereas species that are declining in their native regions are less likely to successfully naturalize elsewhere. This strongly suggests that global naturalization success and the changes of species' native distributions are driven, at least in part, by similar processes.

## Widespread and expanding species naturalize more widely

Species with high occupancies in their native regions decades ago (i.e., in the early period) were more likely to naturalize and to do so in many regions around the globe. This could reflect that such common species were more likely to encountered, transported and introduced elsewhere[22,25]. It could also reflect that the selective pressures that have made certain species common in their native regions also have pre-adapted these species for success as invaders. In order to be able to occur at many locations in its native region, a species has to be able to coexist with a variety of biotas and persist in a wide range of climatic and other environmental conditions[42]. Therefore, species with high occupancies in their native regions are likely to be ecologically versatile, which should also increase the likelihood of naturalization when introduced elsewhere[25,43]. Furthermore, species that are common in their native regions usually have high dispersal abilities[26,44] and are capable of autonomous self-fertilization[45], which are characteristics that also facilitate spread within regions where they are not native[35,46]. For most of the 3920 species in our datasets, information on these traits is not available, so we could not test their importance. However, as woodiness of species is indicative of their growth forms as well as habitat affiliation and is available for all species in our 10 datasets, we ran additional analyses for this variable (see Supplementary Methods for details). While woodiness was not significantly associated with early occupancy in any of the 10 native regions (Table S22), it was positively associated with occupancy change in six native regions (Table S23). The latter might reflect that many woody species are very long lived and therefore might, in contrast to non-woody species, still occur in grid cells even when the environment is no longer optimal. Inclusion of woodiness in the hurdle models slightly increased the explained variation in global naturalization success (Table S24), but did not change the overall conclusions regarding the effects of early occupancy and occupancy change (compare Table 2 and Table S1). When woody species did naturalize, they did so in fewer regions than non-woody species, and this was significant for eight of the 10 datasets (Tables S1, S2–S11). Similarly, Dong et al.[29] recently showed that among introduced plants in China, woody species were less likely to naturalize than non-woody species. Woody species, particularly trees, have longer juvenile periods and/or are probably introduced in lower numbers than non-woody species, and therefore require more time before they spread into multiple regions. Whatever the exact reason, the additional analyses including woodiness indicate that not all of the characteristics associated with species that increase in their native regions are also associated with species that have become widely naturalized globally.

Numerous studies have analyzed how naturalization relates to static measures of commonness in species' native distributions, such as native range size and grid-cell occupancy e.g.[21,34,47]. However, we are not aware of any previous study that also considered temporal changes in measures of commonness within native regions. In line with the idea that the drivers of expansion in native regions and naturalization elsewhere should be largely the same, we found that the global naturalization success of species—measured either as naturalization incidence or extent—was positively associated with occupancy change in nine of the 10 native regions for which we had data (Fig. 2). Although

these associations were highly significant, and inclusion of occupancy change increased the explained variation in naturalization success (Table S24), the standardized effect estimates were always smaller for the occupancy-change index than for occupancy in the early period (Tables S12-S21). The association between occupancy-change and global naturalization success, however, may be an underestimate because many species may already have started to change in occupancy within their native regions before the early census periods, most of which were in the second half of the 20th century. Additionally, for species that occurred in almost all grid cells of the early period, an increase in occupancy was hardly possible (Fig. S3, Table S25). So, although such common species may also be widely naturalized, they would nevertheless have low occupancy-change values. This would have weakened the strength of the association between global naturalization success and the occupancy-change index. In addition, it might also have reduced the likelihood of detecting synergistic effects between early occupancy and occupancy change, as was found for the Czech Republic, the Netherlands and Switzerland. Therefore, the associations between global naturalization success and native range occupancy change might be even stronger than indicated by our analyses.

Species that have been reported previously to increase in occurrence frequencies within their native regions have been found to be adapted to disturbed and anthropogenic habitats[48], to be strong competitors and to have a preference for nutrient-rich habitats[6,11,27]. Similarly, a higher naturalization success has been reported for species that grow in anthropogenic, nutrient-rich and more productive habitats in their native regions[24,49]. It has also been shown that naturalized and invasive species frequently capitalize more on increases in nutrient availability than less successful alien species[50,51]. The similar characteristics of species that have become more common in their native regions and those that have successfully naturalized elsewhere strongly suggests that similar processes drive both phenomena. The Thiérache region in northern France (Table 2, Fig. 2j, Fig. S1j, S2j) was the only region in which the change in occupancy was not significantly associated with global naturalization success. This might reflect that Thiérache was the smallest region with the fewest number of species, and that the early period was not in the second half of the 20th century, but at the end of the 19th century. Although the latter would better capture the occupancies of species prior to the Great Acceleration (i.e., the period of marked increases in human activity, which started in the mid-20th century), the occupancies in the early period for the Thiérache were coarse estimates based on verbal descriptions of the commonness of the species[10]. Consequently, the data for Thiérache were less precise than for the other nine regions, and the analysis had less statistical power because of the fewer number of species.

The relative consistency between the Bernoulli and zero-truncated count parts of the hurdle models suggests that species with high early occupancies and occupancy-change values were both more likely to naturalize and to do so in many regions. Furthermore, with the exception of Thiérache, our results were also consistent across the native regions, despite the large variation in time periods and intervals covered by the datasets (Table 1). This indicates that our results are robust. For Great Britain and Ireland, the original data sources actually provided occupancy data for three different periods (see Supplementary Methods for details). However, irrespective of which period was assigned as an early or later period for calculating the occupancy-change index, the results were largely similar (Tables S26, S27, S29, S30). For the Netherlands, we extracted occupancy data using two different years of split. In Fig. 2 and Table 2, we present the data for the periods before and after 1990, but if we instead used data for the periods before and after 2000, the results were generally the same (Table S32). Furthermore, when we added woodiness and its interactions with early occupancy and occupancy change to the

models, the results were, with a few deviations, largely the same (Tables S1, S2–S11). The main deviations were for Flanders and Germany, in which the positive main effect of occupancy change on the likelihood of being naturalized was no longer significant, but the interaction between occupancy change and woodiness was significant (Table S5, S6). This suggests that for these two native regions the positive effect of occupancy change on the likelihood of naturalization is mainly accounted for by the woody species. Nevertheless, the additional analyses show that the positive associations of global naturalization success with early occupancy and occupancy change in the native regions are robust.

## Characteristics of widespread and expanding species

Overall, our analyses show that species with high values of both early occupancy and occupancy change in their native regions are also successful as naturalized species globally. This suggests that data on occupancy and changes therein in the native range could inform invasion risk assessments. However, it would ultimately be interesting to unravel which characteristics distinguish the group of widespread and expanding species from other species in the native range. For most traits, data are available for only a small proportion of the global flora[28]. Indeed, a recent analysis showed that there are only 10 traits with data available for more than 50 percent of naturalized species[52]. As mentioned above, a notable exception is woodiness, which we therefore also included in an alternative set of hurdle models (Tables S1 and S2–11). Still, as previous studies found that expanding species are typically strong competitors that take advantage of additional resources, we ran additional analyses using Grime's CSR strategy[53] and Ellenberg environmental indicator values[54] for the subsets of species for which these data were available (see Supplementary Methods for details). We found that the competitor scores of the species with high values of both early occupancy and occupancy change were significantly higher than those of all other species in nine of the 10 native regions (Fig. S4, Table S33), whereas the stress-tolerator and ruderal scores were frequently lower (Fig. S5, S6, Table S33). Furthermore, widespread and expanding species had significantly lower Ellenberg indicator values for light (Fig. S7, Table S34), and higher Ellenberg indicator values for nutrients (Fig. S8, Table S34), in all 10 native regions. On the other hand, indicator values for moisture (Fig. S9, Table S34) and temperature (Fig. S10, Table S34) differed significantly between the two groups of species only in some of the native regions, and not in a consistent pattern. Overall, these supplementary analyses align with the findings of previous studies that competitively strong species preferring nutrient-rich habitats tend to be widespread and have recently further increased in occurrence frequencies within their native regions[6,11,27].

## Study limitations

A strength of our study is that we had multiple datasets on temporal changes in the occupancy of native species. However, the data also comes with limitations that should be kept in mind when interpreting the results. First, we only found suitable data for calculating the occupancy-change index for regions in central and north-western Europe. This means that we cannot generalize our findings to species that are native to other continents or to their whole native range. On the other hand, given that Europe is one of the major donors of naturalized plants globally[55], the anthropogenic changes that drive the naturalization of European plant species in other continents are likely to similarly affect the occupancy change of native species in these other continents. As many regions around the globe now have plant distribution atlas data e.g.[56,57], (https://anpsa.org.au/, https://plants.usda.gov/, https://data.canadensys.net/), future reassessments of these distributions will allow for the calculation of occupancy changes in these regions. Second, there might be biases in the recording of different species and in the intensity of recording in different parts of the same region. Moreover, it could be that some species are increasing in part of their native distribution, for example in high-latitude regions, but decreasing in other parts of their native distribution, for example in low-latitude regions. Nevertheless, for species present in multiple datasets, the occupancy-change indices were generally positively correlated between the regions (of the 44 pairwise Pearson correlation coefficients, 41 were significantly positive and only two were significantly negative; Fig. S11). This indicates that the species that are increasing or decreasing in one of their native regions usually also do so in the other ones.

Rabinowitz[36] proposed three dimensions of rarity and commonness —geographic range, habitat specificity and local abundance. Here, we only considered occupancy within native regions, which is arguably only one component of Rabinowitz's geographic range dimension. However, Crisfield et al.[37] recently proposed to add occupancy as another dimension and to remove habitat specificity as a dimension, because the latter is rather a cause of rarity. Native geographic range size is unlikely to have dramatically changed for most species, at least not when quantified as the number of regions (mostly countries) in which a species is native. In our 10 datasets, early occupancy was always positively correlated with native range size (Pearson $r = 0.182$-$0.404$, all $P < 0.001$, Table S35). Local abundance, on the other hand, is likely to have changed for many species in the last decades due to anthropogenic environmental change. The local abundance of species is usually positively related to the other measures of a species' distribution e.g.[58,59]. Previous studies across various spatial scales and taxonomic groups have also shown that species increasing their occupancy are also very likely to increase in abundance at the sites where they occur[60,61]. However, quantitative data for changes in local abundances for large numbers of species in areas comparable in size to our 10 datasets are rare. A notable exception is a recent study by Jandt et al. [62], who analyzed data on changes in the local abundances (cover) of plant species in vegetation plots in Germany over the period 1927–2020. The change in local abundance was positively correlated with the occupancy-change index that we calculated for the native species in Germany (Pearson $r = 0.185$, $n = 1214$, $P < 0.001$). This suggests that, overall, species that have become more widespread in Germany have also become locally more abundant. Thus, when more such data become available from resurveys of vegetation plots[63], it will be worthwhile to look at the interactive effect of changes across different dimensions of commonness in future studies.

In conclusion, our analyses provide strong evidence that many plant species that are spreading as naturalized aliens around the globe also have high occupancies and/or are increasing in occupancy in their native regions. These findings have several major implications. First, if both phenomena are largely driven by anthropogenic environmental changes and the species' characteristics that preadapt them to these changes, this could explain why studies that compared widely naturalized aliens with widespread native species did not find differences in e.g., their responses to nutrient addition[64], soil heterogeneity[65] and competitive abilities[66]. Second, although the objective of our study was not to build a predictive model of drivers of plant naturalization, our findings suggest that measures of commonness and changes therein in the native regions could provide important insights into the likelihood that these species may naturalize after introduction elsewhere. While national assessment schemes for potential invasion risk, such as the Australian weed risk assessment[67], consider whether the assessed alien species is known to be naturalized or invasive in other regions, these risk assessment schemes do not yet consider commonness and its dynamics in the species' native regions. Considering these factors might help policymakers and managers enhance the accuracy and effectiveness of invasion risk assessments, ultimately leading to more informed and proactive conservation strategies.

## Methods

### Datasets of occupancy change in native regions over time

As a metric of commonness within species' native regions, we used occupancy (i.e., the proportion of grid cells across a region in which a species has been recorded). To obtain datasets on native species' occupancies for multiple distinct periods, we searched the scientific literature and online plant atlases and contacted curators of national plant species distribution databases. We found such data for 10 European regions (countries or parts thereof): Austria (number of native species: $n = 2419$), the Czech Republic ($n = 1834$), southeastern Denmark ($n = 921$), Flanders (including the capital region of Brussels) in Belgium ($n = 861$), Germany ($n = 1715$), Great Britain (including the Isle of Man and the Channel Islands) ($n = 1355$), Ireland (including the Republic of Ireland and Northern Ireland) ($n = 910$), the Netherlands ($n = 1115$), Switzerland ($n = 2307$) and the Thiérache region in northern France ($n = 775$) (Fig. 1, Table 1). Across the 10 regions, there were 3920 unique taxa, with 288 occurring in all 10 regions, 1261 present in only a single region, and the remaining ones shared across combinations of two to nine regions (Table S36). Although some are subspecies or varieties, we refer to all taxa as species for simplicity.

For each native species in the 10 datasets (i.e., native regions), we extracted the number of grid cells where it has been recorded in two or more distinct time periods, where each period usually lasted multiple years or decades (Table 1). Most of the data sources provide data for only two time periods, limiting us to the temporal splits available in the original datasets. However, for Great Britain and Ireland, the data sources provide data for three time periods. In addition, for the Netherlands, we could extract grid-cell numbers from the original data source choosing different years of split. We chose 1990 and 2000, as these years were also used by some of the other datasets. To see how robust the results were with regard to the chosen year of split, we analyzed the data for Great Britain, Ireland and the Netherlands using multiple splits.

The regions varied in the number of species, the early and later periods, the durations of these periods and the interval between the periods, as well as in the size and total number of grid cells (Table 1). For example, for the Thiérache region, which had data on 775 native species and their occurrences in 129 grid cells of $4 \times 4$ km, the early and later periods were 1880–1900 and 1957–2005, whereas for Germany, which had data on 1715 native species and their occurrences in 12,024 grid cells of $5' \times 3'$, the early and later periods were 1960–1987 and 1997–2017. For most of the native regions, the data were actual grid-cell frequency counts, and they covered the entire region. However, for the Thiérache region, the grid-cell frequencies for the early period were only coarse estimates based on verbal descriptions of how widespread the species were during that period[10]. Furthermore, for southeastern Denmark, the data do not cover one contiguous region but 11 subregions that do not all border one another (Fig. 1). The data source for southeastern Denmark[7] provides regional abundance data for each of the 11 subregions, where each species' abundance in a subregion was calculated by dividing the number of grid cells occupied by the species by the total number of grid cells for the subregion. From these data, we back-calculated the numbers of grid cells occupied by a species in each subregion, and then combined them across the 11 subregions to get one single occupancy value. Further details on the data for each of the 10 native regions are provided in the Supplementary Methods.

### Species selection and taxonomic harmonization

To select native plant species for inclusion in the final datasets and to harmonize the taxonomic names, we applied one common workflow to all 10 datasets. First, we excluded species that, according to the original data source, are not native to the respective region. When the native status was not provided or not entirely clear, we checked the native status of the species in the corresponding region in the Plants of the World Online database (POWO; https://powo.science.kew.org/ accessed in April 2023). Second, we harmonized the species names according to the taxonomic backbone of the World Checklist of Vascular Plants (WCVP version 11, which is integrated in POWO). This was done in R version 4.2.3[68], initially with the *TNRS* package version 0.3.3[69], and later, after it became available, the *rWCVP* package version 1.0.3[70]. Species that did not have exact matching names in WCVP version 11 or that had multiple matching names were checked manually, and corrections were made when necessary. Species that did not match an accepted name were removed from the datasets. If multiple species in the original dataset were assigned to the same accepted name in WCVP, we kept the one that had the largest number of occupied grid cells. We could not merge the grid cell data because most datasets only provide the number of occupied grid cells, and not the identities of the grid cells occupied by each species.

### Index of occupancy change in native regions

To calculate an occupancy-change index, defined as the change in the proportion of grid cells between the two time periods covered by the data, relative to the expected change based on the occupancy in the early period (i.e., early occupancy), we followed the method developed by Telfer et al.[41]. We chose this method because the resulting occupancy-change index corresponds to the residuals of a weighted regression of the logit-transformed occupancy in the later period on the logit-transformed occupancy in the early period. As a consequence, the occupancy-change index is, compared to other change indices, less sensitive to differences in collection effort between the early and later periods. This is because the index does not quantify the absolute change but quantifies how much larger or smaller the magnitude of a species' occupancy change is relative to species with the same early occupancy. Another advantage is that the occupancy-change index is not correlated with early occupancy, and therefore does not cause multicollinearity issues when both are included in the same statistical model. However, like for any other occupancy-change index, we cannot rule out the possibility that the accuracy of recording certain groups of species might have changed for some regions between the periods.

Following the protocol of Telfer et al.[41], we first calculated the proportion of occupied grid cells in a native region for each species in each time period as $(x + 0.5)/(n + 1)$, where $x$ is the number of grid cells in which the species has been recorded and $n$ is the total number of grid cells in the respective region. As recommended by Telfer et al.[36], we added 0.5 to $x$ and 1 to $n$ to avoid proportion values of zero and one. This was necessary because in the second step, the proportion values were logit transformed, and the logit of zero would be undefined and the logit of one would be infinite. Next, we performed a weighted least-squares linear regression of the logit-transformed proportions of occupied grid cells during the later period as a function of the logit-transformed proportion of occupied grid cells during the early period. The weights were equal to the reciprocal of the variance of the logit-transformed proportions. Data visualization using scatterplots showed that species with low occupancy proportions in the early period deviated from the linear relationship. Therefore, as recommended by Telfer et al.[41], we excluded species occupying fewer than five grid cells during the early period. The resulting regression plots are shown in Fig. S3. The values of the occupancy-change index then correspond to the standardized residuals (i.e., the deviations of the logit-transformed proportions of grid cells occupied by the species in the later period from the regression line). Therefore, for each early occupancy value, we have a more or less symmetrical distribution of negative and positive occupancy-change index values (i.e., standardized residuals; Fig. S3). It should be noted that a positive value of the change index does not necessarily mean that the occupancy of a species has increased between the two periods. Instead, it means that the species occupied a higher proportion of grid cells in the later period than expected based on its occupancy in the early period.

## Data on global naturalization success

To quantify global naturalization success of each species in the 10 native region datasets, we used the Global Naturalized Alien Flora (GloNAF) database. GloNAF is the most comprehensive compendium of lists of naturalized alien vascular plant species for regions around the globe [18, provisionally accepted] and is continuously being updated. We used version 2.0 (extracted in January 2024), with data for 920 non-overlapping regions around the globe, including both mainland regions and islands (Fig. S12). GloNAF regions mostly follow geopolitical boundaries, including countries, states, provinces and individual islands. The GloNAF regions range in size from 0.045 to 2336618 km² (the median size is 34382.71 km²). Global naturalization of a species was quantified as the number of GloNAF regions in which it has naturalized, which is strongly correlated with the cumulative area of these regions[15]. As the species names in GloNAF version 2.0 follow the WCVP taxonomic backbone, the species names in the native occupancy datasets could be directly matched to the GloNAF database.

## Statistical analysis

Because the datasets of the 10 regions differed in many aspects (Table 1), we analyzed each region separately. The separate analyses also ensure that we only make comparisons of changes in occupancy of species that occurred over the same time-period interval. Global naturalization of a species, quantified as the number of GloNAF regions in which the species is naturalized, was the response variable in our analyses. Because large proportions of the species in the 10 datasets have not naturalized in any region (median proportion: 0.300, range: 0.108–0.443), the count data are zero inflated. Therefore, we analyzed global naturalization success as a combination of naturalization incidence (naturalized vs. not naturalized) and naturalization extent (the number of regions where naturalized) using a hurdle model[71] with the *hurdle* function of the *pscl* package version 1.5.5[71] in R. The hurdle model consisted of a generalized linear model (GLM) with a Bernoulli distribution with the logit link function for naturalization incidence (comparing the non-zeros to the zeros), and a GLM with a zero-truncated negative binomial distribution with the log link function for naturalization extent. We used a negative binomial distribution for the zero-truncated count part instead of a Poisson distribution to account for overdispersion. All the statistical tests in the hurdle model were two-sided. In both parts of the model, we included occupancy in the early time period (i.e., early occupancy; EO) the occupancy-change index (i.e., OC) and their interaction as predictor variables. Early occupancy and its interaction with the occupancy-change index were included in the model to test whether species that were widespread within the native regions decades ago were more likely to become naturalized elsewhere, and whether this is particularly true when these widespread species have further increased in native region occupancy. To illustrate whether the relationship between global naturalization success and occupancy change varied depending on the occupancy in the early period, we fitted the predicted relationship between naturalization success (as a combination of the Bernoulli and zero-truncated count part) and the occupancy-change index for the median value of early occupancy, as well as for the 1/6th quantile (i.e., species that were relatively rare in the early period) and 5/6th quantile (i.e., species that were relatively common in the early period; Fig. 2, Fig. S1, S2). To facilitate interpretation and comparison of model estimates across the 10 native regions, early occupancy was scaled to a mean of zero and a standard deviation of one[72]. This was not necessary for the occupancy-change index because it corresponds to standardized residuals, which are already scaled to a mean of zero and a standard deviation of one.

## Reporting summary

Further information on research design is available in the Nature Portfolio Reporting Summary linked to this article.

## Data availability

All the datasets used in this study have been deposited in the Figshare database under the https://doi.org/10.6084/m9.figshare.25487209[73]. Source data are provided with this paper.

## Code availability

The R code used for the statistical analysis is available in Code Ocean with the identifier https://doi.org/10.24433/CO.1618280.v1 and in the Figshare database under the identifier https://doi.org/10.6084/m9.figshare.25487209[73].

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

## Acknowledgements

R.P., N.L.K., A.D. and M.v.K. acknowledge funding from the German Research Foundation DFG (grant numbers 264740629 and 432253815 to MvK). R.P. and W.Z. acknowledges support of the International Max Planck Research School for Quantitative Behavior, Ecology and Evolution (IMPRS-QBEE). W.Z. acknowledges the funding from China Scholarship Council (grant no. 202106100035). M.C. and J.D. were supported by the Czech Science Foundation (project no. 19-28491X). K.G. and W.Y.G. acknowledge funding from the Natural Science Foundation of China (grant no. 32301386 to K.G. and 32171588 and 32471676 to W.Y.G.). P.P. and J.P. were supported by EXPRO grant no. 19-28807X (Czech Science Foundation) and, together with J.D., Z.K. and J.W., by long-term research development project RVO 67985939 (Czech Academy of Sciences). F.E. acknowledges funding from the Austrian Science Foundation FWF (Global 458 Plant Invasions, grant no. I 5825-B). H.K. acknowledges funding of research unit FOR2716 DynaCom (379417748) and Biodiversa+ BioMonI (533271599) from the German Research Foundation (DFG). M.W. acknowledges funding from the German Research Foundation (via iDiv, FZT 118, 202548816). We thank Fang-Wei Lin for help with data extraction. This Open Access publication was supported by the Publication Fund of the University of Konstanz.

## Author contributions

M.v.K. designed research; R.P. performed research; M.v.K & R.P. compiled the data for calculating native range occupancy change; H.V.C., M.C., J.D., G.D., L. E-S., K.G., W.Y. G., Z.K., S.P., J.W., W.D., F.E., H.K., J. P., P.P., M.W. & M.v.K. contributed data; R.P., T.S.F., N.L.K., A.D., W.Z. & M.v.K. prepared the data; R. P. & M.v.K analyzed data; R.P., M.v.K. and T.S.F. wrote the first paper draft and all other coauthors contributed to writing.

## Funding

## Competing interests

All the authors declare no competing interests.

## Additional information

[1]Ecology, Department of Biology, University of Konstanz, Konstanz, Germany. [2]International Max Planck Research School for Quantitative Behaviour, Ecology and Evolution (IMPRS-QBEE), Max Planck Institute of Animal Behaviour, Konstanz, Germany. [3]Department of Biology, University of Puerto Rico – Rio Piedras, San Juan, Puerto Rico. [4]Biometry, Methodology and Quality Assurance, Research Institute for Nature and Forest, Brussel, Belgium. [5]Department of Botany and Zoology, Faculty of Science, Masaryk University, Brno, Czech Republic. [6]Department of Taxonomy, Institute of Botany, Czech Academy of Sciences, Průhonice, Czech Republic. [7]Ecologie et Dynamique des Systèmes anthropisés (UMR CNRS 7058 EDYSAN), University of Picardie Jules Verne, Amiens, France. [8]Department of Botany and Biodiversity Research, University of Vienna, Vienna, Austria. [9]State Key Laboratory of Black Soils Conservation and Utilization, Northeast Institute of Geography and Agroecology, Chinese Academy of Sciences, Changchun, China. [10]Zhejiang Tiantong Forest Ecosystem National Observation and Research Station, Institute of Eco-Chongming, Research Center for Global Change and Ecological Forecasting, School of Ecological and Environmental Sciences, East China Normal University, Shanghai, China. [11]Zhejiang Zhoushan Island Ecosystem Observation and Research Station, Zhoushan, China. [12]State Key Laboratory of Estuarine and Coastal Research, East China Normal University, Shanghai, China. [13]Department of Botany, Faculty of Science, Charles University, Prague, Czech Republic. [14]Department of Agricultural and Environmental Sciences (DiSAA), University of Milan, Milan, Italy. [15]Department of Geoecology, Institute of Botany, Czech Academy of Sciences, Průhonice, Czech Republic. [16]Department of Evolution, Ecology and Behaviour, Institute of Infection, Veterinary and Ecological Sciences, University of Liverpool, Liverpool, UK. [17]Centre for Invasion Biology, Department of Botany and Zoology, Stellenbosch University, Stellenbosch, South Africa. [18]Biodiversity, Macroecology and Biogeography, University of Göttingen, Göttingen, Germany.

[19]Centre of Biodiversity and Sustainable Land Use (CBL), University of Göttingen, Göttingen, Germany. [20]Campus-Institut Data Science (CIDAS), University of Göttingen, Göttingen, Germany. [21]Department of Invasion Ecology, Czech Academy of Sciences, Institute of Botany, Průhonice, Czech Republic. [22]Department of Ecology, Faculty of Science, Charles University, Prague, Czech Republic. [23]German Centre for Integrative Biodiversity Research (iDiv) Halle-Jena-Leipzig, Leipzig, Germany. [24]Zhejiang Provincial Key Laboratory of Plant Evolutionary Ecology and Conservation, Taizhou University, Taizhou, China.
✉ e-mail: rashmi.paudel@uni-konstanz.de

