## [Transparent Peer review file · Nature Communications]

Many plants naturalized as aliens abroad have also become more common within their native regions

Corresponding Author: Ms Rashmi Paudel

Version 0:

Reviewer comments:

Reviewer #1

(Remarks to the Author)

This paper addresses a highly novel question, namely whether species that are successful as non-native species have also become more successful within their native range. I am familiar with the invasions literature and I am unaware of any previous study that has addressed this question. The analyses themselves are conducted soundly and the work is clearly described. I have no major concerns about this manuscript. I do have one suggestion that I believe should be addressed in some way, as well as several additional thoughts that could add additional interpretation to their findings. I detail these considerations below.

The one issue that I believe needs to be better addressed has to do with their choices about where to split the native range data, temporally, i.e., on deciding on a specific year to split “historic” from “recent” distributions with a region. A few of their data sets have natural splits between these periods, e.g., Denmark (with historic data from the 1800s and modern data from less than a decade ago), but more than half of their data sets appear to have been arbitrarily split at 1990 or 2000, e.g., the Netherlands data set is split as before 1990 and after 1990. At a minimum the manuscript should be revised to explain the rationale for these splits. Why are some split at 1990 and why others at 2000 or even, in one case, at 1971. I’m guessing they have some reasonable reason for this, but this isn’t explained and so it makes the reader feel unsure about these decisions. They mention one data set where they look at multiple possible splits, but I really would have liked to have seen an analysis with a few more of these data sets that considered if the year of the split (e.g., 1980 vs. 1990 vs. 2000) ends up impacting the patterns observed. I imagine that their results are pretty robust to the precise year of splits, but I’m not sure. I think addressing this in some way would be useful, even if was just to show this with 2 or 3 more of their study regions. If it proves impractical to do such analyses then I would still expect to see a short paragraph added (or perhaps 2-3 sentences added to an existing paragraph) to the discussion that addresses this concern.

This is super minor, but on line 64 they state that approximately 25% of the planet’s species are threatened with extinction. This seems a little high to be well supported. I would like to see the number revised and or the addition of citations to support this claim.

This study is likely to have an artifactual bias against detecting change in occupancy of species that were historically abundant. Note that this makes their key finding “harder” to detect, i.e., they found their result in spite of the fact that there’s a bias making this result less likely to be observed. This is worth discussing, particularly in the context of explaining why they find a stronger connection between historic occupancy and naturalized success vs. change in occupancy and naturalized success. The issue is that any species that was historically common, i.e., found in most grid cells within a region, cannot add many new grid cells, since they already occupy most of them. This consideration will vary among regions depending upon how fully the common species occupy the entire set of grid cells. So, if the most common species only occupies 40% of the grid cells in the historic period than this consideration won’t be too important; alternatively if the most common species historically occupied 90% of the grid cells (and could only at most occupy the final 10%) than this consideration will be very important. Looking at Fig. S1. I would guess that a region like Austria had common species that occupied almost all of the grid cells historically, which would explain the bow downward in the most common species in the plot, whereas Great Britain, probably did not. In any case, regardless of my exact speculations, I think mentioning this would probably be worthwhile.

While certainly not essential, it would be nice to know if the common species, which became more common through time in

the native range, belong to some easily identifiable group. I can imagine, for example, that most of these species might be agricultural weeds, or perhaps pasture species, etc. Any easy to run analyses that could help elucidate this would provide additional context to the work.

I was a bit surprised the manuscript doesn't mention the "distribution-abundance" relationship, particularly surrounding the discussion of change in local abundance. The distribution-abundance relationship is well described in Jim Brown's book of Macroecology and there are a number of journal articles published on that topic in the 80s and 90s. Basically the pattern (which seems to be pretty robust to taxonomic groups and spatial scales) is that there is a positive correlation between frequency of occurrence and average abundance at sites of occurrence. So, species found at lots of sites tend to also be more abundant at those sites, whereas species found at few sites tend to also have low abundance where they are found. There is disagreement about why this happens, but not really disagreement that it happens. Sometimes there's a more triangular (less linear) relationship, such that some species found at few sites are abundant where they are found, but what never happens is having species found at lots of places that on average are found in low abundance where they do occur. What this means, relative to this paper, is that the species that have increased in frequency are also very likely to have increased in abundance as well.

Dov Sax

(Remarks on code availability)
not applicable

Reviewer #2

(Remarks to the Author)

The aggregated dataset is impressive. The long history of plant records and surveys in Europe allows a unique gathering of information on plant species occupancy and the authors investigate long-lasting questions on the association between species commonness and their ability to naturalize globally. The research questions are relevant to invasion ecology and insights into these dynamics could have important implications to allow managers to target efforts of early detection and control. I also reviewed the code and analysis for Austria, and I am extremely satisfied with the documentation provided and the conciseness of it – great job! With that said, I bring below several important overall topics to consideration:

Given (i) the constraints in including all 10 countries/regions in a single model, the authors' solution by considering them as independent "datasets" modeling each region separately although many share political boundaries, (ii) the low variability explained by the models, and (iii) the narrow geographical scope of the sources regions of species to establish elsewhere, I am having a hard time to be convinced that early occupancy and occupancy change index are driving forces of the number of global regions with naturalization for a given species, and the broad generalizations brought in the title (as if all naturalized plants across the globe follow a similar pattern) and manuscript. Interestingly, the authors bring such caution themselves in L208-209, and it reinforces that the generalization brought up in this manuscript can be unrealistic.

My next big concerns rely on the predictors chosen to be in the model, and not exploration effect sizes (only significance was reported). Given that early occupancy is used to calculate the occupancy change index, one would expect a high correlation among them as well as a lack of independence, producing confounded effects of each variable on the global naturalization metrics (naturalized vs not naturalized; and number of global regions with naturalization). How correlated the predictors are? How can you justify the use of both variables in the same model while interpreting their effects? With respect to effect size and model interpretation, hurdle model effect sizes need to be back-transformed to be interpreted adequately. However, the values presented in Table 2 are values directly extracted from the model outputs.

Across all models, the predictors explained a low portion of the variability within the response variable (13% to 28%). I wonder if the effects of early occupancy and the occupancy change index would have a strong effect (dependent on the model) if other important variables were included, such as predictors that account for human influence (such as Global Human Modification; Kennedy, C. M., Oakleaf, J. R., Theobald, D. M., Baruch-Mordo, S., & Kiesecker, J. (2018). Global human modification. <https://doi.org/10.6084/m9.figshare.7283087>, figshare), productivity (net primary productivity – NPP), and other environmental variables that have been shown to strongly influence the establishment of non-native plants across the globe. Therefore, I would recommend incorporating other drivers of the establishment (such as the ones I have just mentioned) in the model to allow a more accurate understanding of the relative importance of the different forms of occupancy (if both can actually be in a single model) investigated in the manuscript.

Overall, the consistency in terminology used can be largely improved. Doing so, will allow the readers to follow the thought process and story of the paper more easily, as well as connect the different components across the sections. Here are my recommendations: (1) the words 'countries', 'native range', and 'regions' are used interchangeably, when they shouldn't. I recommend if using regions, to clarify in their first appearance that they correspond to the entirety of their respective country in X number of cases. I would prefer to not read 'native range' in the context of the paper as be synonym of a region, given that, as pointed out in the paper (figure S6) there are several species that are native to multiple countries/regions (side note: the number of common species across countries/regions should be added to the methods section). (2) I strongly suggest the revision of the terminology associated with referring to non-native species as aliens. I suggest the authors revise the discussion provided here (<https://doi.org/10.1111/brv.13071>) and here (<https://doi.org/10.1002/fee.2561>), and consider a complete switch to 'non-native', including in the title. (3) the use of "at home" and "winners at home" is confusing, and it is not consistently used across the manuscript. Particularly when the "winners at home" are associated with the distribution of

species within a political boundary (such as country) rather than its native range (which more frequently than not does not follow such boundaries). Additionally, by referring as “whether species that are increasing at home” [L99] implies that the authors directly measured direct expansion in species range, which wasn’t the case and was explained in the methods on L3314-L317. (4) the interchangeable use of “changes in native-range commonness” and “occupancy change index” makes it hard to follow.

Legends of figures: I would suggest the revision of the legend of all figures and tables to be completely understandable in isolation. For example, in table 2 the model results show results for both “naturalized or not” and for the “number of regions where naturalized”, not only the later as directed. The understanding of hurdle models can be not trivial, so the more detailed explanation of what exactly the Bernoulli and zero-truncated count parts means biologically is strongly encouraged. Another example, Figure S6 legend is quite uninformative, and can only be understood after reading the main text. I pointed out here a couple examples, but all legends should be revised for clarity. These are suggestions that I believe will improve the accurate understanding of your paper and its biological implication.

Early occupancy was not clearly explained anywhere in the main manuscript or supplement. How was this data specifically obtained given the earliest period of data available for a country/region? And more generally, how occupancy was defined? Let’s say there is a period of 3 years and in y1 a species was present in 6 out of 10 cells; y2, in 4 cells being 2 new ones compared to y1; y3 in 7 cells being one new one. What is the occupancy of this species? Given the disparity of sources and formats the data was obtained in the first place for each country/region, was the count of cells to obtain occupancy consistent across the country/region? How?

I believe that the abstract does not contain enough information to understand how the knowledge gap was analyzed, so I suggest a thorough revision. For example, in L45-L46, “particularly common plant species” where? The abstract presents the results from the truncated negative binomial portion of the model only, therefore defining what ‘global naturalization’ means in L46-47 is necessary. Also, a basic ecological interpretation/definition of “occupancy-change index” and “early period occupancy” would allow a larger audience to benefit from these findings. The latter was not formally defined in the paper, and it hampers the understanding of the choice of approach and the findings, as I mentioned in another piece of this review. Lastly, the abstract would gain a larger audience by having a closing statement with the implication for management and policy-related actions for species that follow the pattern described in this manuscript.

Overall structure of the paper: In my first read of the main paper, I would argue that the content necessary to fully understand the analysis and research findings isn’t present. The authors should keep in mind that the methods section only appears at the end of the paper, so other sections should provide enough information about the methods and definitions of terms used that allow the readers to accurately understand the results and discussion.

How similar are the boundaries of the 10 focal regions compared to the same regions from GloNAF?

Specific comments and recommendations:

- I am curious to understand why “declining species” was a relevant keyword selected by the authors.
- L84: add citation to support statement.
- L88: a good example in which countries and ranges are used interchangeably. Clarifying language would improve the understanding and cohesion of this paper.
- L104: the statement “largely the same species” has dubious meaning, clarify.
- L105: the statement “two (or more) periods” has no context up to this point and hinders the reader from grasping the methods used in the paper in general terms. Also, to the best of my understanding of this paper, for each country/region, each main model shown in the main manuscript contains only the comparison between two time periods. Adding “or more” is confusing and might better if brought up later when nuances in the results are explained.
- All content presented in the supplement should be referenced in the main text.
- L106: “for each native species in each region” how about species that are present in more than one region? There is some mention of them when Figure S6 is referred to, but given the proximity of each of the countries/regions used in this study some short explanation on why region rather than native range extrapolating political boundaries were chosen.
- L107: “occupancy-change index” hasn’t been explained and needs clarification.
- L109-110: “the number of regions where the species has become naturalized” this text is confusing. It seems that one is referring to the 10 regions, but in fact, it is from the regions defined by GloNAF to defined where a species is naturalized or not.
- L110: “early period” is mentioned in the abstract and here, but the term hasn’t been described yet.
- L121-L122: this statement contradicts statements on L314-317. These species are more common than what would be expected given their occupancy in the early period.
- L125-L126: with respect to “the zero-truncated count part of the hurdle model, not with the Bernoulli part”, I suggest changing the language similar to above: the likelihood of being naturalized outside native range vs naturalized in more regions at the global scale.
- L131: to convey the message more clearly, it might be helpful to associate the occupancy-change index as a metric of commonness early on in the text and use this term across the paper.
- L136: this statement “increased in occupancy” is unclear. Since when?
- L145-148: statements seem repeated from the introduction. I suggested the authors be mindful of which information is essential and necessary in the discussion by focusing on an in-depth discussion of the results.
- L149-150: what is the environmental variability within the native range studied per species? is the set of species used in this study representative of species distributed in a wide range of environmental conditions?
- L152-154: are these traits present in the focal species studied here? Could such information be pulled from global

databases, such as TRY, and incorporated here?

- L186-187: if the early period chosen had a specific reasoning for it, it should be clarified in the introduction, not in the discussion.

- L202-204: I cannot follow the connection of this sentence to the previous after. "richer-get-richer" is a term generally used in the context of species richness, and it doesn't seem to apply here, but also the directionality of the negative is the opposite in the sentence? Consider revising and defining "Matthew effect" and or "richer-get-richer".

- L223: how were "losers" quantified in this paper? Similar language also shows up in the keywords, and it makes it confusing.

- L228-229: why habitat specificity might have not changed?

- L245-251: this discussion is really interesting and I would suggest to be the driving force of the importance of this study and findings (if changes in the methods are performed and conclusions hold) - to aid ways in which findings from core science can be transferred to managers and policymakers.

(Remarks on code availability)

the results for "Austria" are reproducible - from analysis to figures. According to the authors and my understanding of their methods, each model was run the same way, only the input data for other regions (data available on figshare) changes.

Reviewer #3

(Remarks to the Author)

The study aims to compare whether the changes in select plant species' native range size correlates with naturalization history elsewhere. The authors examine changes in grid occupancy drawing on historic grid-based surveys of the vascular flora of 10 areas in Europe. The premise is interesting and the results, though correlative, make intuitive sense. However, there are some methodological concerns that render the conclusion that plants that have naturalized as aliens abroad have also become more common in their native range unearned.

As the authors note, there are many definitions of commonality, and arguably the one being used in this study is not truly a metric of such. At best, the authors are examining a proxy of range size. The authors do a good job of mentioning potential limitations – e.g., one cannot know how abundant these species are within each grid cell. Further, it is likely that the surveys upon which the study relies were conducted with varying protocols and specific objectives, certainly across regions and possibly even over time. Even if the surveys were more or less standardized, the analyses conducted here can produce misleading results for species that have been recorded more efficiently/accurately in one survey than the others.

More importantly, the assayed regions do not represent the full native range of the species examined. It is possible for any given species that while, say, in the Czech Republic its range has increased, the range may have decreased in other parts of its native range. Thus, it is unknown how a species' entire native range size has changed. Though the authors claim that the general positive correlation between occupancy-change indices they observed among regions for the same species indicates that native range 'winners' and 'losers' were generally consistent across regions, these regions may constitute a small or biased sampling of these species' entire native ranges. Also, many of the species that have naturalized elsewhere are known to have fairly large native ranges that extend well beyond the regions included in the study. Further, it seems that only 44 species (among thousands) were present in multiple regions. Along these lines, without more context regarding the full native ranges of these species it is difficult to say whether the results of this study, though suggestive, truly provide strong evidence that many plant species that are spreading as naturalized aliens around the globe also have high occupancies or are increasing in occupancy in their native range.

GloNAF regions are used as the unit of analyses regarding naturalized ranges. However, the manuscript never mentions what these regions are. How were they defined? Based on the map provided in the supplements, it seems that these regions largely follow geopolitical boundaries and come in many shapes and sizes. Thus, the number of regions where a species has become naturalized may not be a suitable metric for the degree of global naturalization (and comparison with occupancy in a limited sampling of these species' native ranges).

The methods require more detail. Pertinent information is often relegated to the supplements or other publications. For instance, the authors should note how many species were examined in the main text so that the numbers they present and discuss can be understood in context. While it is fine to refer to methods in previous studies, there should at least be enough information in the manuscript for the reader to be able to interpret the results. For instance, it was necessary to refer back to Telfer (2002) repeatedly to understand the figures presented in the results as well as some of the choices made in the study. Along these lines, the grid counts for each region were not done the same way. For instance, in Denmark, the number of occupied grid cells of each species in each region was calculated by multiplying the regional abundance of each species by the total number of referenced grid cells for each region, combining the grid-cell data for each taxon across the 11 regions of the country that were surveyed to get one single value. In contrast, simple grid cell counts were used in some other regions. While there is no reason not to use available data, this makes the results from each region (which were modeled separately) less comparable, and I recommend that the authors harmonize the approach to grid cell counting and present those results as well.

Additional comments:

The 'winners' and 'losers' angle seems a bit colloquial.

Table 1 is not referenced in the text correctly.

Fig S6 – x and y are not defined.

Unscale the occupancy index when presenting and discussing the results – the scaled values are confusing to interpret and the index is already ultrametric.

The layout of Fig.2 is a bit confusing with the inset with multiple abbreviations and asterisks. Also, why not just replace this figure with the corresponding one in the supplements that show the raw data as noted in the caption (S5)?

Where were the 1/6 and 5/6 quantiles chosen for illustration?

As GloNAF data are not freely accessible, please present information on which specific regions each species was naturalized in.

Are the identities of the grid cells occupied by each species known? If so, perhaps the potential drivers of range change could be examined.

The “Gridcells_earlyperiod” and “Gridcells_laterperiod” values for Thiérache are not whole numbers— according to the metadata these should be simple counts of the number of cells occupied by each species and the supplementary methods do not state any special circumstances (unlike Germany or Denmark).

(Remarks on code availability)

I did not try running the code but it looked reasonable.

Version 1:

Reviewer comments:

Reviewer #1

(Remarks to the Author)

The authors have addressed all of my concerns.

(Remarks on code availability)

Reviewer #2

(Remarks to the Author)

I appreciate the careful point-by-point revisions given in response to my previous comments and concerns. I appreciate the thorough revision of the text to improve the consistency of terms used—the main text is now a lot easier to digest. Moving forward, I have two broad, major comments that are detailed below. Then, those are followed by minor comments:

1) I mentioned this before and I noticed that Reviewer 3 also made a comment on this: using the regions as units of native ranges, particularly because they follow political boundaries. So, could you provide the supplement with how many species are present across all regions? How many species are present in all regions but one? And so forth? And how many species are uniquely present in a single region?

2) Woodiness was added in complementary models as a way to address some of the concerns of Reviewer Dov Sax. If this inclusion is kept, the main text would need some introduction of the relatedness of woodiness, range expansion, and probability of becoming naturalized, so, therefore, justifying why this a reasonable trait to be evaluated. Lines L171-173 do not justify sufficiently. I would pull as base of some of the arguments and justification given in the rebuttal.

Minor comments:

Results: The tables with the main results of your models (i.e., mention of tables S14-S30 should potentially show up earlier, as it is the results of the main models (without woodiness)

L91-93: The statement is somewhat confusing. Also, should it be “increasing their occupancy” rather than “increasing” only in L92? Please, revise

L103: do you mean “range over time”?

L199-200: would you be able to provide either a table or a simple graph per region with how many species out of the total per region were present in all its grid cells? This would help the reader to be aware of how many species, from the poll studies, have this pattern.

L228: “...species with high early...”?

(Remarks on code availability)

I reviewed the code during the first round of revisions.

Reviewer #4

(Remarks to the Author)

This is a broad analysis stating that certain plant species, likely due to a combination of human actions and the plant features, are favored under human uses of the landscape in their ranges and in new regions. The novelty relies on including change in abundance/occupancy in the native range as a predictor of naturalization somewhere else. These findings could be used to generate list of species that could become invasive, lists that could be compared with already generated watch lists to assess how much information this predictor is adding.

Since changes in occupancy/abundance were not analyzed as a function of other drivers than time, e.g., human activities targeting particular habitats/plant communities, results might very indirectly be assessing the causes of naturalization, because increase in occurrence and naturalization might have to do with habitats selected rather than intrinsic features of the plant species. This is a point brought by the reviewers that has been dismissed by the authors (see next comment). Analyzing what it made those species increase in their native and introduced ranges would be of greater consequence for management and conservation.

Much more informative would have been to analyze features of the species that were already abundant and that increased their occurrences, e.g., are ruderal species over-represented in that group? What are the main traits characterizing those species? That again would provide more useful information for management and conservation. Maybe the data is not available for all, but it could be done for the ones that it is. Line 207-216 make that point. Many studies are using global trait data sets to do it, if authors have decided not to do it that's their choice but claiming that the reason for not doing it is that that information is not available is not accurate.

It looks like early occupancy, and to some extent woodiness, are a better predictor of naturalization than change in occupancy, Table S3, this should be emphasized more in the main text because data on changes overtime may not be available, but actual native range and woodiness are, and those could be used on their own as predictors. Especially considering that change in occupancy is being predicted by early occupancy. Also, adding information on how much variance was additionally accounted for by including change in occupancy, vs a model with early occupancy and woodiness only, would help to understand the scope of this predictor.

My suggestion would be to change 'region' for countries. Region usually indicates ecological, e.g., climatic, differences, but these data sets reflect political units. Sorting the data into actual ecological regions might have shown more interesting/informative results.

It is not clear if changes in occupancy overtime were standardized by the period of time between census, i.e., more change expected if longer time between censuses.

I can't tell the difference between Table S3 and the ones that follow for each country, information is the same.

(Remarks on code availability)

The analyses are done using standard techniques and standard software packages.

Version 2:

Reviewer comments:

Reviewer #2

(Remarks to the Author)

I have now revisited the authors' responses to the inquiries raised by Reviewer #4 and find that they have made appropriate revisions and/or provided clear and well-justified explanations, both in the rebuttal and in the manuscript, for not implementing certain suggestions. I am satisfied with the current version of the manuscript.

As Reviewer #2, I also appreciate the authors' thorough and clear responses to my own comments. Thank you for the careful attention to detail in your point-by-point reply.

Reviewer #2

(Remarks on code availability)

I reviewed the code during the first round of revisions.

Dear reviewers,

We have now revised the manuscript entitled “Plants that have naturalized as aliens abroad have also become more common at home during the Anthropocene” (“Many plants naturalized as aliens abroad have also become more common within their native regions” in revised version), based on the helpful and constructive comments. Below we provide our point-by-point responses to the comments. Please, note that the line numbers refer to the clean version of the manuscript with highlights and our responses are in blue.

Sincerely,

Rashmi Paudel (on behalf of all authors)

Reviewer #1 (Remarks to the Author):

This paper addresses a highly novel question, namely whether species that are successful as non-native species have also become more successful within their native range. I am familiar with the invasions literature and I am unaware of any previous study that has addressed this question. The analyses themselves are conducted soundly and the work is clearly described. I have no major concerns about this manuscript. I do have one suggestion that I believe should be addressed in some way, as well as several additional thoughts that could add additional interpretation to their findings. I detail these considerations below.

RESPONSE: We thank the reviewer (Dov Sax) for his positive and constructive comments.

The one issue that I believe needs to be better addressed has to do with their choices about where to split the native range data, temporally, i.e., on deciding on a specific year to split “historic” from “recent” distributions with a region. A few of their data sets have natural splits between these periods, e.g., Denmark (with historic data from the 1800s and modern data from less than a decade ago), but more than half of their data sets appear to have been arbitrarily split at 1990 or 2000, e.g., the Netherlands data set is split as before 1990 and after 1990. At a minimum the manuscript should be revised to explain the rationale for these splits. Why are some split at 1990 and why others at 2000 or even, in one case, at 1971. I’m guessing they have some reasonable reason for this, but this isn’t explained and so it makes the reader feel unsure about these decisions. They mention one data set where they look at multiple possible splits, but I really would have liked to have seen an analysis with a few more of these data sets that considered if the year of the split (e.g., 1980 vs. 1990 vs. 2000) ends up impacting the patterns observed. I imagine that their results are pretty robust to the precise year of splits, but I’m not sure. I think addressing this in some way would be useful, even if was just to show this with 2 or 3 more of their study regions. If it proves impractical to do such analyses then I would still expect to see a short paragraph added (or perhaps 2-3 sentences added to an existing paragraph) to the discussion that addresses this concern.

RESPONSE: We now provide information on the choices of the splits (lines 323-324) as “*Most of the data sources provide data for only two time periods, limiting us to the temporal splits available in the original datasets*”. Most of the datasets provided data for only two time periods and we were therefore limited to the splits chosen in the original datasets. For the Netherlands, however, we can also extract the grid-cell numbers using different years of the split, therefore, to see how robust the results are with regard to the chosen split 1990, we now also extracted data using the year 2000 as split. We chose 2000 because it was also the year of split for some of the other datasets. Like we already found for Great Britain and Ireland, the patterns for the Netherlands were very similar for the different years of the split. We added these results to the Supplement (TableS30).

This is super minor, but on line 64 they state that approximately 25% of the planet’s species are threatened with extinction. This seems a little high to be well supported. I would like to see the number revised and or the addition of citations to support this claim.

RESPONSE: We now specify that the 25% applies to the animal and plant species that have been assessed, and we refer to the IPBES 2019 report (lines 64-65).

This study is likely to have an artifactual bias against detecting change in occupancy of species that were historically abundant. Note that this makes their key finding “harder” to detect, i.e., they found their result in spite of the fact that there’s a bias making this result less likely to be observed. This is worth discussing, particularly in the context of explaining why they find a stronger connection between historic occupancy and naturalized success vs. change in occupancy and naturalized success. The issue is that any species that was historically common, i.e., found in most grid cells within a region, cannot add many new grid cells, since they already occupy most of them. This consideration will vary among regions depending upon how fully the common species occupy the entire set of grid cells. So, if the most common species only occupies 40% of the grid cells in the historic period than this consideration won’t be too important; alternatively, if the most common species historically occupied 90% of the grid cells (and could only at most occupy the final 10%) than this consideration will be very important. Looking at Fig. S1. I would guess that a region like Austria had common species that occupied almost all of the grid cells historically, which would explain the bow downward in the most common species in the plot, whereas Great Britain, probably did not. In any case, regardless of my exact speculations, I think mentioning this would probably be worthwhile.

RESPONSE: We now discuss this consideration in lines 199-205.

While certainly not essential, it would be nice to know if the common species, which became more common through time in the native range, belong to some easily identifiable group. I can imagine, for example, that most of these species might be agricultural weeds, or perhaps pasture species, etc. Any easy to run analyses that could help elucidate this would provide additional context to the work.

RESPONSE: We agree that, although it would not be essential for answering our research question, adding such information would be very interesting. Unfortunately, habitat information is currently not available for all 3920 species in our datasets. Instead, we now added information on woodiness of the species (where we assume that woody species are typical for forest habitats and non-woody species for open habitats). We found that while woody species had higher occupancy-change values than non-woody species in six of the 10 native regions (Table S2), woody species were less likely to become widely naturalized. Nevertheless, the positive association between naturalization and occupancy change remained largely unaffected by inclusion of woodiness as an additional variable in the hurdle models. This shows that our results are robust. We added the results of the additional analyses in the Supplements (Tables S3, S4-13.) and refer to them in the Discussion (lines 173-185 & lines 238-245).

I was a bit surprised the manuscript doesn't mention the "distribution-abundance" relationship, particularly surrounding the discussion of change in local abundance. The distribution-abundance relationship is well described in Jim Brown's book of Macroecology and there are a number of journal articles published on that topic in the 80s and 90s. Basically the pattern (which seems to be pretty robust to taxonomic groups and spatial scales) is that there is a positive correlation between frequency of occurrence and average abundance at sites of occurrence. So, species found at lots of sites tend to also be more abundant at those sites, whereas species found at few sites tend to also have low abundance where they are found. There is disagreement about why this happens, but not really disagreement that it happens. Sometimes there's a more triangular (less linear) relationship, such that some species found at few sites are abundant where they are found, but what never happens is having species found at lots of places that on average are found in low abundance where they do occur. What this means, relative to this paper, is that the species that have increased in frequency are also very likely to have increased in abundance as well.

RESPONSE: We thank the reviewer (Dov Sax) for pointing this out, and we now mention the "distribution-abundance" relationship in the Discussion as "*The local abundance of species is usually positively related to the other measures of a species' distribution [e.g. 50, 51]. Previous studies across various spatial scales and taxonomic groups, have also shown that species increasing their occupancy are also very likely to increase in abundance at the sites where they occur [e.g. 52, 53].*" (lines 278-281) and refer to Jim Brown's book in the Introduction (line 98, reference number 34).

Dov Sax

Reviewer #1 (Remarks on code availability):

not applicable

Reviewer #2 (Remarks to the Author):

The aggregated dataset is impressive. The long history of plant records and surveys in Europe allows a unique gathering of information on plant species occupancy and the authors investigate long-lasting questions on the association between species commonness and their ability to naturalize globally. The research questions are relevant to invasion ecology and insights into these dynamics could have important implications to allow managers to target efforts of early detection and control. I also reviewed the code and analysis for Austria, and I am extremely satisfied with the documentation provided and the conciseness of it – great job! With that said, I bring below several important overall topics to consideration:

RESPONSE: We thank the reviewer for the compliments and the constructive comments.

Given (i) the constraints in including all 10 countries/regions in a single model, the authors' solution by considering them as independent “datasets” modeling each region separately although many share political boundaries, (ii) the low variability explained by the models, and (iii) the narrow geographical scope of the sources regions of species to establish elsewhere, I am having a hard time to be convinced that early occupancy and occupancy change index are driving forces of the number of global regions with naturalization for a given species, and the broad generalizations brought in the title (as if all naturalized plants across the globe follow a similar pattern) and manuscript. Interestingly, the authors bring such caution themselves in L208-209, and it reinforces that the generalization brought up in this manuscript can be unrealistic.

RESPONSE: We did not intend to say that all plants that have naturalized have also become more common at home. We have now carefully checked the text and toned down any broad generalizations that are not directly supported by our data. For example, to make it clear that not all plants that have naturalized have become more common at home, we now write “Many plants ...” Instead of just “Plants ...”. Although we used early occupancy and the occupancy-change index as predictors in the models, and naturalization as response variable, we did not imply that early occupancy and the occupancy-change index are the driving forces of the number of global regions with naturalization for a given species. Instead, we believe that our results suggest that both commonness at home and naturalization elsewhere are correlated because they share similar drivers. We now mention this more explicitly (lines 102-104 & 156-157).

My next big concerns rely on the predictors chosen to be in the model, and not exploration effect sizes (only significance was reported). Given that early occupancy is used to calculate the occupancy change index, one would expect a high correlation among them as well as a lack of independence, producing confounded effects of each variable on the global naturalization metrics (naturalized vs not naturalized; and number of global regions with naturalization). How correlated the predictors are? How can you justify the use of both variables in the same model while interpreting their effects?

RESPONSE: We can understand the reviewer’s concern about possible correlations between early occupancy and the occupancy-change index. However, the main reason why we chose to use the

approach by Telfer et al. (2002) for calculating the occupancy-change index is that this approach avoids such a correlation. The occupancy-change index corresponds to the residuals of the regression of logit-transformed latter occupancy on logit-transformed early occupancy. So, for each early occupancy value, we have a symmetrical distribution of negative and positive occupancy-change index values (i.e. residuals), as can be seen in Fig. S4. We now explain this more clearly in the manuscript (lines 400-402). We also mention at the end of the Introduction (lines 116-117) that the occupancy-change index is not correlated with initial occupancy. Nevertheless, we now also calculated Pearson correlation coefficients between early occupancy and the occupancy-change index, and they were indeed low:

Region	Pearson correlation between early occupancy and occupancy change index	P value
Austria	-0.042	<0.05
Czech Republic	0.040	0.087
Denmark	-0.005	0.871
Flanders	-0.026	0.454
Germany	0.036	0.137
Great Britain	0.176	<0.01
Ireland	0.063	0.057
The Netherlands	0.016	0.595
Switzerland	0.003	0.889
Thiérache	-0.014	0.703

Regarding the effect sizes, in addition to Table 2, which just provides a broad overview, we have also provided the full model results, including model estimates along with their respective standard errors and significance value, in supplementary Tables S14-S23. In these models, we scaled the early occupancy to mean of zero and a standard deviation of one, this was not necessary for the occupancy-change index, as it corresponds to standardized residuals, which are already scaled to a mean of zero and a standard deviation of one. Therefore, we believe that since the estimates are based on standardized values, the interpretation of the model coefficients (as effect sizes) is straightforward.

With respect to effect size and model interpretation, hurdle model effect sizes need to be back-transformed to be interpreted adequately. However, the values presented in Table 2 are values directly extracted from the model outputs.

RESPONSE: Indeed, in Table 2, we provide the model estimates of the Bernoulli and zero-truncated count parts of the hurdle models. This table mainly serves as an overview of which effects were significant in the 10 different regions. The full models' outputs are provided in Table S14-S23. We are not entirely sure how the reviewer would like to see these estimates back

transformed. For the Bernoulli part, we could use the model estimates to calculate odds ratios, but for the zero-truncated count part, there is to our knowledge no equivalent. However, Figure 2 shows, based on the coefficients of both parts of the hurdle model, the predicted relationships for the number of regions where a species is naturalized. So, we believe that what the reviewer wants is already visualized in this figure. In addition, the predicted relationships for the Bernoulli and zero-truncated count parts separately are shown in Figures S1 and S2.

Across all models, the predictors explained a low portion of the variability within the response variable (13% to 28%). I wonder if the effects of early occupancy and the occupancy change index would have a strong effect (dependent on the model) if other important variables were included, such as predictors that account for human influence (such as Global Human Modification; Kennedy, C. M., Oakleaf, J. R., Theobald, D. M., Baruch-Mordo, S., & Kiesecker, J. (2018). Global human modification. <https://doi.org/10.6084/m9.figshare.7283087>, figshare), productivity (net primary productivity – NPP), and other environmental variables that have been shown to strongly influence the establishment of non-native plants across the globe. Therefore, I would recommend incorporating other drivers of the establishment (such as the ones I have just mentioned) in the model to allow a more accurate understanding of the relative importance of the different forms of occupancy (if both can actually be in a single model) investigated in the manuscript.

RESPONSE: Given that many factors known to affect the naturalization process, we believe that 13-28% is not a low portion of the variability for this kind of studies. Nevertheless, we fully agree that other predictors may explain more of the variation in global naturalization success, and some of authors of this study have done such analyses. However, the aim of the current analyses was not to build a predictive model for global naturalization success. Instead, our objective was to assess whether most of the species that have started to expand outside their native ranges have also expanded within their native ranges. Therefore, although we believe that the inclusion of other predictors will increase the proportion of explained variance, it will not provide additional insights into our research question. Moreover, the variables suggested by the reviewer (i.e. human modification and NPP) are characteristics of spatial units (e.g. grid cells or regions). Therefore, such variables are useful for explaining spatial patterns in naturalized plant richness. In our study, however, the units of analysis are species, and our measures of naturalization success are not spatially explicit (i.e. our models predict naturalization incidence and number of regions, but not *where* a species is naturalized). Adding these variables to our analysis would therefore not be possible, and we do not believe that they would help to answer our research question or change the current results. However, based on this comment and on a comment of Reviewer #1, we now performed additional analyses in which we included woodiness, a trait that is available for most species, as another species-level “predictor” in the model. The results for the association of naturalization success with the change in occurrence frequency remained largely the same, and the portion of variation explained by the models only slightly increased (the R2 values range from 14 to 28 percent). These additional analyses have been added to the Supplements.

Overall, the consistency in terminology used can be largely improved. Doing so, will allow the readers to follow the thought process and story of the paper more easily, as well as connect the different components across the sections. Here are my recommendations: (1) the words 'countries', 'native range', and 'regions' are used interchangeably, when they shouldn't. I recommend if using regions, to clarify in their first appearance that they correspond to the entirety of their respective country in X number of cases. I would prefer to not read 'native range' in the context of the paper as be synonym of a region, given that, as pointed out in the paper (figure S6) there are several species that are native to multiple countries/regions (side note: the number of common species across countries/regions should be added to the methods section). (2) I strongly suggest the revision of the terminology associated with referring to non-native species as aliens.

I suggest the authors revise the discussion provided here (<https://doi.org/10.1111/brv.13071>) and here (<https://doi.org/10.1002/fee.2561>), and consider a complete switch to 'non-native', including in the title. (3) the use of "at home" and "winners at home" is confusing, and it is not consistently used across the manuscript. Particularly when the "winners at home" are associated with the distribution of species within a political boundary (such as country) rather than its native range (which more frequently than not does not follow such boundaries). Additionally, by referring as "whether species that are increasing at home" [L99] implies that the authors directly measured direct expansion in species range, which wasn't the case and was explained in the methods on L3314-L317. (4) the interchangeable use of "changes in native-range commonness" and "occupancy change index" makes it hard to follow.

RESPONSE: We thank the reviewer for spotting the inconsistencies in terminology and for the constructive suggestions. We now use consistent terminology throughout the manuscript.

(1) When referring to the regions, we now consistently use 'regions' (or 'native regions') instead of 'countries' or 'native range'. However, we still use 'native range' when we really mean the entire native range of a species. In response to the side note mentioned by the reviewer under point 1, we have now added the number of common species across the native regions to Figure S6.

(2) We prefer to keep the term 'alien' throughout the manuscript for several reasons. First, 'alien' is the term recommended by the widely used invasion stages frameworks of Richardson et al. (2000, Diversity and Distributions 6:93-107, DOI:10.1046/j.1472-4642.2000.00083.x) and Blackburn et al. (2011, Trends in Ecology and Evolution 26:333-339, doi:10.1016/j.tree.2011.03.023), and it also the term used in the recent IPBES assessment ("Thematic Assessment Report on Invasive Alien Species and their Control"; <https://www.ipbes.net/ias>), a work that has set the standards for the current invasion research. So, using another term, in our opinion, will not tame the terminological tempest in invasion science. Furthermore, we cannot avoid the term 'alien' because it is part of the name of the main database (the Global Naturalized Alien Flora; GloNAF) used for the data on naturalization success. We have published many papers that use the GloNAF database, and we consistently used the term "alien". Thus, for the sake of consistency of our own work, we prefer to continue using the same term. Second, we are aware that the term 'alien' has unfortunately become politicized, particularly in North America. However, when we have to come up with a new term each time a

term becomes politicized, the terminological chaos in invasion biology will increase rather than decrease. In this regard, it should also be noted that the term 'non-native' is also considered harmful by some researchers (see the database of harmful terminology in ecology and evolution, <https://www.eeblanguageproject.com/repository>, associated with Cheng et al. (2023, Trends in Ecology and Evolution 5:381-384, <https://doi.org/10.1016/j.tree.2022.12.011>). Having said this, as much as we prefer the term 'alien', if the editor insists on us replacing 'alien' with another term, we are willing to do so.

(3) We now avoid the term 'at home' and instead use 'in their native regions'.

(4) We used "changes in native-range commonness" only once in the manuscript (line 160 of the previous version), and there we do not want to change it to "occupancy change index". This is because we were not referring exclusively to the index that we used but to any change in commonness. However, we now replaced it with "temporal changes in measures of commonness within native regions" (line 189).

Legends of figures: I would suggest the revision of the legend of all figures and tables to be completely understandable in isolation. For example, in table 2 the model results show results for both "naturalized or not" and for the "number of regions where naturalized", not only the later as directed. The understanding of hurdle models can be not trivial, so the more detailed explanation of what exactly the Bernoulli and zero-truncated count parts means biologically is strongly encouraged. Another example, Figure S6 legend is quite uninformative, and can only be understood after reading the main text. I pointed out here a couple examples, but all legends should be revised for clarity. These are suggestions that I believe will improve the accurate understanding of your paper and its biological implication.

RESPONSE: We now have thoroughly revised all legends, including Table 2 and Figure S6, which is now Figure S3 (lines 659-672 & lines 147-150 in supplementary).

Early occupancy was not clearly explained anywhere in the main manuscript or supplement. How was this data specifically obtained given the earliest period of data available for a country/region? And more generally, how occupancy was defined? Let's say there is a period of 3 years and in y1 a species was present in 6 out of 10 cells; y2, in 4 cells being 2 new ones compared to y1; y3 in 7 cells being one new one. What is the occupancy of this species? Given the disparity of sources and formats the data was obtained in the first place for each country/region, was the count of cells to obtain occupancy consistent across the country/region? How?

RESPONSE: We have now moved the definition of "occupancy" and "early occupancy" to the end of the Introduction as "*Here, we test the hypothesis that many of the plant species that have become widely naturalized across the globe are also increasing in occupancy (i.e. the proportion of grid cells across a region in which a species has been recorded) within their native regions. To test this hypothesis, one would ideally have time series data of grid-cell occupancies for the entire native ranges of species. However, as such data is not available, we instead retrieved data on grid-cell occupancies of vascular plant species during an early period (i.e. early occupancy) and a later period, each usually covering multiple years, for 10 regions in Europe, which are referred as*

native regions (Fig. 1)." (lines 106-107 & 110). We also defined early occupancy again in Method section (line 371).

Most datasets only provide the number of grid cells where a particular species is present without specifying in which grid cells the species occurs. So, if a species was observed in 10 grid cells in the first period and in 10 grid cells in the second period, these could be the same grid cells, but they could also be totally different grid cells. Furthermore, each period with occupancy data was multiple years, and occupancy was then the number of grid cells in which a species has been observed during that multi-year period. So, when the period was three years (as in the example of the reviewer), we only have data on the number of grid cells in which a species has been observed during these three years, irrespective of whether it was in the first, second or third year or any combination of those. In other words, we have data on the total number of grid cells for the period, which usually lasted multiple years, but not for year 1, year 2 and year 3 separately. Additionally, we only have information on the total number of grid cell for each period but not their identities. We now mention this explicitly in lines 111, 322-323 & 364-366.

As mentioned in the manuscript, the datasets for the different regions varied with regard to the early and later periods, the durations of these periods, and the number and sizes of the grid cells. However, within a region and period, the count of cells to obtain occupancy was consistent across species, at least for nine of the ten datasets. Therefore, and because the occupancy calculated for a particular region was relative to the total number of grid cells of that region, we believe our approach ensures comparability across regions. The only exception was the Thiérache region, where in the early period the number of grid cells was an approximation obtained by converting ordinal rarity classes that were used in the late 19th century to a mean number of occupied grid cells per rarity class. This might, as discussed in lines 217-226, be one of the reasons why the results of the Thiérache region deviate from those of the others.

I believe that the abstract does not contain enough information to understand how the knowledge gap was analyzed, so I suggest a thorough revision. For example, in L45-L46, "particularly common plant species" where? The abstract presents the results from the truncated negative binomial portion of the model only, therefore defining what 'global naturalization' means in L46-47 is necessary. Also, a basic ecological interpretation/definition of "occupancy-change index" and "early period occupancy" would allow a larger audience to benefit from these findings. The latter was not formally defined in the paper, and it hampers the understanding of the choice of approach and the findings, as I mentioned in another piece of this review. Lastly, the abstract would gain a larger audience by having a closing statement with the implication for management and policy-related actions for species that follow the pattern described in this manuscript.

RESPONSE: We have now thoroughly revised the Abstract and believe that we have addressed all major points made by the reviewer, despite the strict limit of 150 words.

Overall structure of the paper: In my first read of the main paper, I would argue that the content necessary to fully understand the analysis and research findings isn't present. The authors should keep in mind that the methods section only appears at the end of the paper, so other sections should provide enough information about the methods and definitions of terms used that allow the readers to accurately understand the results and discussion.

RESPONSE: We are fully aware that the Methods section is placed at the end of the article and have therefore now provided a concise overview of our approach at the end of the Introduction (lines 105-124). We also double checked whether there were any cases where definitions of terms were included only in the Methods. When that was the case, we moved them into the earlier sections.

How similar are the boundaries of the 10 focal regions compared to the same regions from GloNAF?

RESPONSE: Five of the focal regions (the Netherlands, Flanders, Germany, Switzerland and the Czech Republic) have exactly matching GloNAF regions. For the focal regions Austria, Ireland and Great Britain, GloNAF has data for the different subregions (e.g. England, Scotland and Wales for Great Britain). For the focal regions Denmark (southeast) and Thiérache, there are no exactly matching regions in GloNAF, but GloNAF has data for the corresponding countries (Denmark and France). We have not added this information to the manuscript, as we do not see why it would matter whether the focal regions match GloNAF regions. However, if the editor wants us to add this information, we will be happy to do so.

Specific comments and recommendations:

- I am curious to understand why "declining species" was a relevant keyword selected by the authors.

RESPONSE: We tried to include keywords that are not yet used in the Title or Abstract, so that search engines would have a higher chance to find our paper. We included "declining species" because this term is frequently used for species that used to be common but have become rare recently. As this applies to many of the species that have a negative occupancy-change index, we think that the term is appropriate. While we chose to frame the perspective of our manuscript (e.g., in the title or abstract) in terms of species that have increased in commonness, the finding that species that are declining in their native range are less likely to be naturalized elsewhere is an equally valid and interesting extension of our results. We now mention this explicitly in the Discussion section (lines 155-156). A search for "declining species" in Web of Science (12 June 2024) gave 747 results. Therefore, we consider that the use of this keyword will help people with an interest in declining species to find our manuscript.

- L84: add citation to support statement.

RESPONSE: We have now added a citation (line 86).

- L88: a good example in which countries and ranges are used interchangeably. Clarifying language would improve the understanding and cohesion of this paper.

RESPONSE: We now used the standardized term “region” to improve cohesion (lines 92).

- L104: the statement “largely the same species” has dubious meaning, clarify.

RESPONSE: We rephrased the sentence as “*Here, we test the hypothesis that many of the plant species that have become widely naturalized across the globe are also increasing in occupancy (i.e. the proportion of grid cells across a region in which a species has been recorded) within their native regions*” (lines 105-107).

- L105: the statement “two (or more) periods” has no context up to this point and hinders the reader from grasping the methods used in the paper in general terms. Also, to the best of my understanding of this paper, for each country/region, each main model shown in the main manuscript contains only the comparison between two time periods. Adding “or more” is confusing and might better if brought up later when nuances in the results are explained.

RESPONSE: We now removed “or more” from this sentence and also rephrased the sentence (line 109-112).

- All content presented in the supplement should be referenced in the main text.

RESPONSE: All content presented in the supplement is now referenced in the main text. Note, however, that we do not directly refer to Table S26 and Table S29 in the main text but that we refer to these tables in the elaborate descriptions of the individual datasets in the Supplementary Methods (to which we refer in the main text).

- L106: “for each native species in each region” how about species that are present in more than one region? There is some mention of them when Figure S6 is referred to, but given the proximity of each of the countries/regions used in this study some short explanation on why region rather than native range extrapolating political boundaries were chosen.

RESPONSE: We would have liked to have data on changes in occupancy for the entire native ranges of species. However, as data on grid cell occupancy are only available for some of the regions that are part of the species’ native ranges, this was unfortunately not possible. Therefore, we had to use data for separate regions, and indeed, as mentioned in the manuscript, some species occur in multiple data sets. We now mention already at the end of the Introduction the reason for analyzing regions rather than native range (lines 107-112) and the reason for analyzing each region separately (lines 123-124).

- L107: “occupancy-change index” hasn’t been explained and needs clarification.

RESPONSE: We now added an explanation in lines 112-116 of the Introduction and in lines 369-372 of the Methods section.

- L109-110: “the number of regions where the species has become naturalized” this text is confusing. It seems that one is referring to the 10 regions, but in fact, it is from the regions defined by GloNAF to defined where a species is naturalized or not.

RESPONSE: We now rephrased the sentence as *“We then used hurdle models to analyse how global naturalization success — a combination of naturalization incidence (i.e. whether or not a species has become naturalized, which can be modelled by using a Bernoulli distribution) and naturalization extent (i.e. the number of regions where a naturalized species has become naturalized, which can be modelled using a zero-truncated negative binomial distribution) — correlates with occupancy in the early period and the occupancy change within the species’ native regions.”* (lines 117-123).

- L110: “early period” is mentioned in the abstract and here, but the term hasn't been described yet.

RESPONSE: We now make it clear earlier in this paragraph of the Introduction that there were two periods, an early period and a later period (lines 110-111).

- L121-L122: this statement contradicts statements on L314-317. These species are more common than what would expected given their occupancy in the early period.

RESPONSE: We changed the statement in the previous L121-L122 (now lines 135-137) accordingly to *“So, overall, our findings indicate that many species that have increased in occupancy in their native regions more than species with an identical early occupancy, have also become widely naturalized elsewhere in the world”*.

- L125-L126: with respect to “the zero-truncated count part of the hurdle model, not with the Bernoulli part”, I suggest changing the language similar to above: the likelihood of being naturalized outside native range vs naturalized in more regions at the global scale.

RESPONSE: We have now revised the statement (lines 130-132).

- L131: to convey the message more clearly, it might be helpful to associate the occupancy-change index as a metric of commonness early on in the text and use this term across the paper.

RESPONSE: We thank the reviewer for this suggestion. However, the occupancy change index is a metric of the change in commonness, not a metric of commonness per se. As commonness has multiple dimensions, we would prefer not to replace occupancy-change index with the commonness-change index throughout the text.

- L136: this statement “increased in occupancy” is unclear. Since when?

RESPONSE: The increase in occupancy has been observed since the period for which we have early occupancy data across regions. We have rephrased the sentence as *“but also for species that have since then increased in occupancy within their native regions (Table 2).”* to clarify this (lines 149-150).

- L145-148: statements seem repeated from the introduction. I suggested the authors be mindful of which information is essential and necessary in the discussion by focusing on an in-depth discussion of the results.

RESPONSE: As we think that it is more important to mention these possible explanations in the Discussion than in the Introduction, we have now removed the similar statement from the Introduction.

- L149-150: what is the environmental variability within the native range studied per species? is the set of species used in this study representative of species distributed in a wide range of environmental conditions?

RESPONSE: For each of the 10 focal regions, the datasets include the vast majority of plant species that are native there. So, the datasets should include both species that occur in a wide range of environmental conditions and species that occur in only a subset of environmental conditions. However, as we only have data on the number of grid cells occupied by a species and no data on which grid cells they actually occur in, we do not have measures of the environmental variability each species experiences within the native region.

- L152-154: are these traits present in the focal species studied here? Could such information be pulled from global databases, such as TRY, and incorporated here?

RESPONSE: We agree that an analysis using trait data would be nice, but even in global databases such as TRY the number of species with data for specific traits is still very poor. This also applies to data on dispersal abilities and autonomous self-fertilization. The largest dataset on autonomous self-fertilization was compiled by Razanajatovo et al. (2016; Nature Communications 7:13313), and included only 1752 of the ca. 350,000 species globally. However, in additional analyses, we now considered the woodiness of the species. We chose woodiness because it is indicative of both growth form and habitat type (i.e. regarding whether the species is likely to occur in open [non-woody] or closed [woody] habitats), and because it is available for all species in our study. We found that while woody species had higher occupancy-change values than non-woody species in six of the 10 native regions (Table S2), woody species were less likely to become widely naturalized. Nevertheless, the positive association between naturalization and occupancy change remained largely unaffected by inclusion of woodiness as an additional variable in the hurdle models. This shows that our results are robust. We added the results of the additional analyses in the supplements (Tables S3, S4-S13) and refer to them in the Discussion (lines 171-185 & lines 238-245).

- L186-187: if the early period chosen had a specific reasoning for it, it should be clarified in the introduction, not in the discussion.

RESPONSE: We did not have any specific reasoning for that early period. It was simply the period for which data happened to be available. So, we did not have any *a priori* hypothesis about how the years of the early period would affect the results. Therefore, we believe that the early period of the Thiérache period does not require a clarification in the Introduction.

- L202-204: I cannot follow the connection of this sentence to the previous after. “richer-get-richer” is a term generally used in the context of species richness, and it doesn’t seem to apply here, but also the directionality of the negative is the opposite in the sentence? Consider revising and defining “Matthew effect” and or “richer-get-richer”.

RESPONSE: In principle, it is repeating the preceding sentence saying that the successful species become even more successful. As it is not essential to compare this to the Matthew effect and the richer-get-richer effect, we decided to delete the sentence about these effects.

- L223: how were “losers” quantified in this paper? Similar language also shows up in the keywords, and it makes it confusing.

RESPONSE: “Losers” referred to species that have declined (i.e. have a negative change in occupancy). Based on this comment and a comment by Reviewer #3, we decided to remove the term ‘loser’ (and to be consistent, also the term ‘winner’) from the manuscript to avoid any confusion.

- L228-229: why habitat specificity might have not changed?

RESPONSE: We do not exclude the possibility, as habitat specificity could in principle change through evolution. However, we think it is very unlikely that this will have happened for many species within less than a century. As a new framework on the dimensions of rarity, added occupancy as a dimension, and removed habitat specificity as a dimension (Crisfield et al. 2024, doi: 10.1111/ecog.07037), we decided to remove the text about habitat specificity.

- L245-251: this discussion is really interesting, and I would suggest to be the driving force of the importance of this study and (if changes in the methods are performed and conclusions hold) - to aid ways in which findings from core science can be transferred to managers and policymakers.

RESPONSE: We thank the reviewer for pointing this out. We added a few sentences to emphasize how our research can help managers and policymakers (lines 304-306).

Reviewer #2 (Remarks on code availability):

the results for "Austria" are reproducible - from analysis to figures. According to the authors and my understanding of their methods, each model was run the same way, only the input data for other regions (data available on figshare) changes.

RESPONSE: That is correct.

Reviewer #3 (Remarks to the Author):

The study aims to compare whether the changes in select plant species’ native range size correlates with naturalization history elsewhere. The authors examine changes in grid occupancy

drawing on historic grid-based surveys of the vascular flora of 10 areas in Europe. The premise is interesting and the results, though correlative, make intuitive sense. However, there are some methodological concerns that render the conclusion that plants that have naturalized as aliens abroad have also become more common in their native range unearned.

RESPONSE: We thank the reviewer for the general support of the manuscript and the insightful comments.

As the authors note, there are many definitions of commonality, and arguably the one being used in this study is not truly a metric of such. At best, the authors are examining a proxy of range size. The authors do a good job of mentioning potential limitations – e.g., one cannot know how abundant these species are within each grid cell. Further, it is likely that the surveys upon which the study relies were conducted with varying protocols and specific objectives, certainly across regions and possibly even over time. Even if the surveys were more or less standardized, the analyses conducted here can produce misleading results for species that have been recorded more efficiently/accurately in one survey than the others.

RESPONSE: We agree with the reviewer that there are many definitions of commonness, and we indeed point out in our manuscript that commonness has multiple dimensions. While we cannot say how abundant a species is within a grid cell, we politely disagree with the reviewer that our metric (i.e. the grid-cell occupancy frequency in a region) would not truly be a metric of commonness. We now added a reference to the recent review by Crisfield et al. (2024, *Ecography* e07037, doi: 10.1111/ecog.07037), which explicitly defines occupancy as a dimension of commonness. In fact, the authors proposed to replace the habitat specificity category of Rabinowitz's classification by occupancy. Furthermore, as also pointed out by Reviewer #1, positive abundance-occupancy relationships are commonly found across taxa, and we now mention this in the Discussion (lines 278-281).

We also agree with the reviewer that the different data sets that we used may have varying protocols and objectives, possibly even over time. However, because each region was analysed separately, we do not think that variation in protocols among regions is problematic. Variation in protocols over time within a region is not problematic either, as this is accounted for by the Telfer method that we used to calculate the occupancy-change index. We explain this in the Methods section (lines 372-378). However, we cannot exclude that in some of the regions there have been changes in how accurately certain groups of species were recorded, and we now mention this potential limitation (lines 380-383). Moreover, we would like to emphasize that despite the variation among the 10 datasets in different aspects, our results remain largely consistent across regions, which we believe strengthens our conclusions.

More importantly, the assayed regions do not represent the full native range of the species examined. It is possible for any given species that while, say, in the Czech Republic its range has increased, the range may have decreased in other parts of its native range. Thus, it is unknown how a species' entire native range size has changed. Though the authors claim that the general

positive correlation between occupancy-change indices they observed among regions for the same species indicates that native range ‘winners’ and ‘losers’ were generally consistent across regions, these regions may constitute a small or biased sampling of these species’ entire native ranges. Also, many of the species that have naturalized elsewhere are known to have fairly large native ranges that extend well beyond the regions included in the study. Further, it seems that only 44 species (among thousands) were present in multiple regions. Along these lines, without more context regarding the full native ranges of these species it is difficult to say whether the results of this study, though suggestive, truly provide strong evidence that many plant species that are spreading as naturalized aliens around the globe also have high occupancies or are increasing in occupancy in their native range.

RESPONSE: We appreciate the insightful comments from the reviewer. As acknowledged in our manuscript, one of the limitations of our study is that grid-cell occupancy data (for at least two time periods) is only available for regions in north-western and central Europe. There are more regions with grid-cell occupancy data, but these regions unfortunately do not yet have repeated surveys. Consequently, we paid attention to the fact that we cannot generalize our findings to the entire native range of species (lines 208-209, previous version and lines 251-253 in revised version). However, we do suggest that similar anthropogenic changes may have comparable effects on species occupancy in other native regions, though further studies are required when more data become available.

Regarding the reviewer’s concern about the number of species present in multiple regions, we believe that the reviewer misunderstood our correlations between occupancy change indices. We did not find that only 44 species occur in multiple regions. What we found is that in 44 out of the 45 possible pairwise combinations of regions, the occupancy-change indices of the species that occurred in both regions of a pair were positively correlated. We believe this is pretty strong evidence that species that have increased in occupancy in one native region usually also have increased in occupancy in other native regions, even though we do not have data for the entire native range. In Fig. S6 (Fig. S3 in revised version), these correlations are shown, and we now also added the number of species shared between the regions in each pair. Additionally, we have now toned down throughout the manuscript any sentences in which we made broad generalization of native ranges, including the title.

GloNAF regions are used as the unit of analyses regarding naturalized ranges. However, the manuscript never mentions what these regions are. How were they defined? Based on the map provided in the supplements, it seems that these regions largely follow geopolitical boundaries and come in many shapes and sizes. Thus, the number of regions where a species has become naturalized may not be a suitable metric for the degree of global naturalization (and comparison with occupancy in a limited sampling of these species’ native ranges).

RESPONSE: We thank the reviewer for pointing out that we failed to mention what the GloNAF regions are. The GloNAF database is a compilation of lists of naturalized alien plants for different regions around the globe. The version we used (version 2.0), contains list of naturalized vascular

plant taxa for 920 non-overlapping regions. These regions include both mainland regions and islands, and most of them indeed follow geopolitical boundaries, because flora books and species inventories, which are the main data sources of GloNAF, are mostly done for administrative regions. As a consequence, the regions vary in size from 0.045 to 2336618 km² (median size is 34382.71 km²). Instead of the number of GloNAF regions, one could use the cumulative area of the regions in which they are naturalized, but this is highly correlated with the number of regions (Spearman rho = 0.770), as shown in Pyšek et al. 2017 (*Preslia* 89:203-274, DOI: 10.23855/preslia.2017.203). We now added more detailed information on the GloNAF database in the Methods (lines 410-415).

The methods require more detail. Pertinent information is often relegated to the supplements or other publications. For instance, the authors should note how many species were examined in the main text so that the numbers they present and discuss can be understood in context. While it is fine to refer to methods in previous studies, there should at least be enough information in the manuscript for the reader to be able to interpret the results. For instance, it was necessary to refer back to Telfer (2002) repeatedly to understand the figures presented in the results as well as some of the choices made in the study. Along these lines, the grid counts for each region were not done the same way. For instance, in Denmark, the number of occupied grid cells of each species in each region was calculated by multiplying the regional abundance of each species by the total number of referenced grid cells for each region, combining the grid-cell data for each taxon across the 11 regions of the country that were surveyed to get one single value. In contrast, simple grid cell counts were used in some other regions. While there is no reason not to use available data, this makes the results from each region (which were modeled separately) less comparable, and I recommend that the authors harmonize the approach to grid cell counting and present those results as well.

RESPONSE: The numbers of species in each dataset are provided in Table 1 of the main text. We now also added these numbers in brackets behind each region in lines 313-318. We also moved some information about the regions that was not yet provided in Table 1 of the main text from the Supplements to the main text (lines 321-330). Furthermore, we now provide more detailed information regarding our choices that were based on Telfer (2000) in the 'Index of occupancy change in native regions' subsection of the Methods. In addition, as recommended by Reviewer #2, we also revised all the figures and table captions to make them more informative and understandable in isolation (Fig. 2, Fig. S1- S5, Table 2, Table S1-S31).

Data on changes in grid-cell occupancy are scarce, and there is variation in how these data were collected in the different regions. However, the total count of grid cells was generally consistent across regions. In the case of Denmark, there were 14 subregions, and not all of them were immediately adjacent to the other subregions. The paper that provides the data for Denmark, Nielsen et al. (2019), determined a species' abundance in each subregion by dividing the number of occupied grid cells of each species by the total number of grid cells for the subregion. Since, Nielsen et al. (2019) provide such abundance data only for 11 subregions, out of 14 subregions, we back calculated the number of grid cells occupied by a species in each of the 11 subregions.

Then we combined them across the 11 subregions to get one single occupancy value for each species. This way we made the data more comparable to the data that we have for the other regions, as recommended by the reviewer. We now explain our calculation of grid-cell occupancy for Denmark in more detail in the Methods (lines 340-347).

Additional comments:

The 'winners' and 'losers' angle seems a bit colloquial.

RESPONSE: In response to this comment and a similar comment by one of the other reviewers, we decided to remove these terms from the manuscript.

Table 1 is not referenced in the text correctly.

RESPONSE: While we previously cited Table 1 in the Methods, when describing the 10 focal regions, we have now also included a citation to Table 1 at the end of the Introduction (line 124).

Fig S6 – x and y are not defined.

RESPONSE: We now added the axis labels in Fig. S6 (Fig. S3 in revised version).

Unscale the occupancy index when presenting and discussing the results – the scaled values are confusing to interpret and the index is already ultrametric.

RESPONSE: We thank the reviewer for this comment. We now realized that the unscaled occupancy-change index values are already standardized because they correspond to the standardized residuals of the logit-logit regression used to calculate the index. In other words, the values were already centered to zero with a standard deviation of one prior to scaling them. So, we now use unscaled values for the occupancy-change index.

The layout of Fig.2 is a bit confusing with the inset with multiple abbreviations and asterisks.

Also, why not just replace this figure with the corresponding one in the supplements that show the raw data as noted in the caption (S5)?

RESPONSE: We followed the reviewer's advice and replace Fig. 2 with Fig. S5 of previous version.

Where were the 1/6 and 5/6 quantiles chosen for illustration?

RESPONSE: We chose the 1/6th and 5/6th quantiles to illustrate whether or not the relationship between global naturalization and occupancy change depended on the occupancy in the early period. We wanted to show this relationship not only for the median occupancy in the early period but also for a higher and a lower occupancy in the early period. So, we categorized species according to whether they were relatively common, intermediately common or rare during the earlier time period. To achieve this, we divided the data points into three equally-sized groups and, we show the lines for the centres (medians) of these groups, which are the 1/6th, 3/6th (median) and 5/6th quantiles. We could have chosen more groups, but then the figures would be

very busy. We now added more details on the choice of the quantiles in the Methods (lines 440-446).

As GloNAF data are not freely accessible, please present information on which specific regions each species was naturalized in.

RESPONSE: We now uploaded the data on figshare. Please, also note that the 2019 version of GloNAF has been published as a data paper, and that the most recent data has always been freely shared upon request.

Are the identities of the grid cells occupied by each species known? If so, perhaps the potential drivers of range change could be examined.

RESPONSE: Indeed, it would be very interesting to actually examine the potential drivers of range change. However, for all datasets, we only have the number of grid cells where a particular species is present without specifying the identities of the grid cells. We mention this in lines 364-366.

The “Gridcells_earlyperiod” and “Gridcells_laterperiod” values for Thiérache are not whole numbers— according to the metadata these should be simple counts of the number of cells occupied by each species and the supplementary methods do not state any special circumstances (unlike Germany or Denmark).

RESPONSE: We thank the reviewer for pointing this out. We mentioned the reason for this indirectly in line 187 of the previous version, but we now realize that we did not explain this in full detail in the Methods. For Thiérache, the authors of the original dataset had ‘proper’ grid-cell occupancy data for the latter period (that is why the species have whole numbers for that period in the dataset. However, for the early period, the data were not actual grid-cell frequencies but verbal descriptions of how widespread the species were. These verbal descriptions were converted by the authors into grid-cell occupancies, and this resulted in some numbers that are not whole. We now also moved information about the Thiérache region from the supplementary to the main manuscript (Methods section line 338-340).

Reviewer #3 (Remarks on code availability):

I did not try running the code but it looked reasonable.

Dear Reviewers,

We were pleased to read that Reviewer #1 was happy with the changes that we had made, and we thank Reviewers #2 and #4 for their constructive comments. According to the new suggestions, we have made further revisions to our manuscript entitled “Many plants naturalized as aliens abroad have also become more common within their native regions”. Below we provide point-by-point responses to all comments. Please, note that the line numbers refer to the clean version of the manuscript with highlights and our responses are in blue.

Sincerely,

Rashmi Paudel (on behalf of all authors)

REVIEWER COMMENTS

Reviewer #1 (Remarks to the Author):

The authors have addressed all of my concerns.

RESPONSE: We thank the reviewer for their time, and we are pleased that we have addressed all of their concerns.

Reviewer #2 (Remarks to the Author):

I appreciate the careful point-by-point revisions given in response to my previous comments and concerns. I appreciate the thorough revision of the text to improve the consistency of terms used—the main text is now a lot easier to digest. Moving forward, I have two broad, major comments that are detailed below. Then, those are followed by minor comments:

RESPONSE: We are pleased that the reviewer appreciates the thorough revisions of the text, and we thank them for the constructive comments.

1) I mentioned this before and I noticed that Reviewer 3 also made a comment on this: using the regions as units of native ranges, particularly because they follow political boundaries. So, could you provide the supplement with how many species are present across all regions? How many species are present in all regions but one? And so forth? And how many species are uniquely present in a single region?

RESPONSE: We have now added a table in the Supplements that provides a summary of how many species are present in how many of the 10 regions (Table S36), and we refer to it in the main text (line 392). Among the 3920 unique species across all ten regions, 288 species are present in all regions, and 1261 are present in only one region. Additionally, we have provided an Excel file in the data repository detailing the presence and absence of all 3,920 species across the 10 regions.

# regions where species is present	# species
10	288
9	186
8	144
7	130

6	165
5	218
4	356
3	487
2	685
1	1261

2) Woodiness was added in complementary models as a way to address some of the concerns of Reviewer Dov Sax. If this inclusion is kept, the main text would need some introduction of the relatedness of woodiness, range expansion, and probability of becoming naturalized, so, therefore, justifying why this a reasonable trait to be evaluated. Lines L171-173 do not justify sufficiently. I would pull as base of some of the arguments and justification given in the rebuttal.

RESPONSE: We thank the reviewer for pointing this out, and we now introduce the potential importance of woodiness (and other traits) in the Introduction (Lines 104-117).

Minor comments:

Results: The tables with the main results of your models (i.e., mention of tables S14-S30 should potentially show up earlier, as it is the results of the main models (without woodiness)

RESPONSE: We previously referred to Tables S14-S30 (Tables S12-S21 in the revised version) in the caption of Table 2, which summarizes the estimates and significance of all models. We have now also referred to them when we first mention Table 2 in the Results section (Line 163).

L91-93: The statement is somewhat confusing. Also, should it be “increasing their occupancy” rather than “increasing” only in L92? Please, revise

RESPONSE: We thank the reviewer for spotting this. Indeed, it should be “increasing their occupancy”. We have now revised the statement as “*However, this concept has not been assessed in plants. The findings that naturalized plants and those spreading in their native range share a common set of traits suggests that it may be the case*” (Lines 120-122).

L103: do you mean “range over time”?

RESPONSE: We thank the reviewer for pointing this out, and we have now revised the text accordingly (Line132).

L199-200: would you be able to provide either a table or a simple graph per region with how many species out of the total per region were present in all its grid cells? This would help the reader to be aware of how many species, from the poll studies, have this pattern.

RESPONSE: Across all regions, no single species is present in all grid cells of its respective region. However, some species were present in nearly all grid cells. To illustrate this, we have updated Fig. S4 (Fig. S3 in the revised version) by adding a vertical line representing the logit of the total number of grid cells in each region. This allows for a clear visualization of species that are close to this threshold.

In addition, the following table, which has been added to the supplements (Table S25), and is referred to in line 243, provides the number of species present in at least 95% of the total grid cells per region, during the earlier time period:

Regions	# total grid cells	# species present in all grid cells	# species present in more than 95% grid cells	# total native species used in analysis
Austria	2600	0	2	2419
Czech Republic	2551	0	0	1834
Denmark	263	0	5	921
Flanders	985	0	29	861
Germany	12024	0	65	1715
Great Britain	2852	0	0	1355
Ireland	1007	0	8	910
The Netherlands	1685	0	30	1115
Switzerland	1827	0	0	2307
Thiérache	129	0	0	775

L228: "...species with high early..."?

RESPONSE: Corrected accordingly (Line 271) by adding "with".

Reviewer #2 (Remarks on code availability):

I reviewed the code during the first round of revisions.

Reviewer #4 (Remarks to the Author):

This is a broad analysis stating that certain plant species, likely due to a combination of human actions and the plant features, are favored under human uses of the landscape in their ranges and in new regions. The novelty relies on including change in abundance/occupancy in the native range as a predictor of naturalization somewhere else. These findings could be used to generate list of species that could become invasive, lists that could be compared with already generated watch lists to assess how much information this predictor is adding.

RESPONSE: We thank the reviewer for their compliments, for recognizing the potential application of our research, and for their constructive feedback.

Since changes in occupancy/abundance were not analyzed as a function of other drivers than time, e.g., human activities targeting particular habitats/plant communities, results might very indirectly be assessing the causes of naturalization, because increase in occurrence and naturalization might have to do with habitats selected rather than intrinsic features of the plant species. This is a point brought by the reviewers that has been dismissed by the authors (see next comment). Analyzing what it made those species increase in their native and

introduced ranges would be of greater consequence for management and conservation.

RESPONSE: Our main objective was to assess whether species that are increasing at home are also the ones that are increasing elsewhere. If this is the case, this would mean that information on native occupancy dynamics could inform invasion risk assessments (irrespective of the traits that underlie the occupancy dynamics). We now mention this more explicitly in the manuscript (lines 63-64 & lines 294-295). However, we appreciate the reviewer's insightful observation and agree that multiple factors other than time could influence species occupancy in their native regions and their naturalization success. As we mentioned in our previous response, we do not have information on the actual grid cells in which the species occurred, because most datasets only provide the number of grid cells where a particular species is present without specifying in which grid cells the species occurs. Therefore, we cannot explicitly test the role of habitats in driving occupancy patterns. In addition, although such an analysis would be very interesting, it would go far beyond our objective of testing whether species that have increased in occupancy in their native regions are largely the same ones that have increased globally as naturalized alien species.

We politely disagree that we dismissed the "habitat" comment of the previous reviewer. In response to the similar comment by Dov Sax to the previous version of our manuscript, and as stated in our previous rebuttal, we chose to add information on woodiness because it is indicative of both growth form and habitat type. Where we assume that woody species are likely to occur in closed or forest habitats, whereas non-woody species predominantly occur in open habitats. We believe that ultimately the habitat associations of species will depend on their intrinsic features. Although we cited already several papers that have analysed how changes in native occurrence relate to species features, we have now—in response to this comment and the reviewer's subsequent comment—analysed whether species that were common and have further increased are characterized by certain ecological strategies (Grime's competitor, stress tolerator and ruderal strategies) and by ecological indicator values. For more details, see our response to the next comment.

Much more informative would have been to analyze features of the species that were already abundant and that increased their occurrences, e.g., are ruderal species over-represented in that group? What are the main traits characterizing those species? That again would provide more useful information for management and conservation. Maybe the data is not available for all, but it could be done for the ones that it is. Line 207-216 make that point. Many studies are using global trait data sets to do it, if authors have decided not to do it that's their choice but claiming that the reason for not doing it is that that information is not available is not accurate.

RESPONSE: Even though many studies are using global trait data sets, we would like to reiterate that these data sets are very incomplete. In a recent analysis of trait-data availability for naturalized alien species, it was shown that there are only ten traits with data available for more than 50% of the species (<https://ecoevorxiv.org/repository/view/7852/>), and we have now also included this information in the manuscript (lines 297-299). Woodiness is one of the few traits that is available for most plant species, and therefore we had already included this trait during the previous revision. We believe that the results of studies based on very incomplete trait data should be interpreted with caution (as there might be a bias in the availability of data). Nevertheless, we fully agree that the analyses proposed by the reviewer might be informative. Therefore, we have now added data from a global dataset on

CSR-strategies (available for 2125 of the 3920 species in our data sets) from Guo et al. (2018), and data on Ellenberg ecological indicator values (available for 2893 of the 3920 species from Tichý et al. (2023) and the Pladias database of the Czech Flora and Vegetation; Chytrý et al. (2021)).

We classified species as widespread and expanding if they had an early occupancy higher than the median and if they had a positive occupancy change. We then analysed, in line with the reviewer's suggestion, whether the trait value of this group of species differed from all other species. These additional analyses showed that the group of species that were already abundant and also increased their occurrences, are overall characterized by high values along the competitor axis of Grime's CSR-strategy triangle (in 9 of the 10 regions) and low values along the ruderal and stress-tolerator strategies. Furthermore, this group of species had significantly lower Ellenberg indicator values for light, and higher Ellenberg indicator values for nutrients, in all ten native regions. We present these additional results in the Supplements (Tables S33-S34, Fig. S4- Fig. S10) and refer to them in the Discussion (lines 301-316).

References:

Chytrý M., et al. Pladias Database of the Czech Flora and Vegetation. *Preslia* **93**, 1–87 (2021).

Guo, W.Y., et al., The role of adaptive strategies in plant naturalization. *Ecology letters* **21**(9), 1380-1389 (2018).

It looks like early occupancy, and to some extent woodiness, are a better predictor of naturalization than change in occupancy, Table S3, this should be emphasized more in the main text because data on changes overtime may not be available, but actual native range and woodiness are, and those could be used on their own as predictors. Especially considering that change in occupancy is being predicted by early occupancy. Also, adding information on how much variance was additionally accounted for by including change in occupancy, vs a model with early occupancy and woodiness only, would help to understand the scope of this predictor.

RESPONSE: We thank the reviewer for this suggestion. We have now emphasized Table S3 (Table S1 in the revised version) more strongly in the main manuscript (lines 156-161, lines 215-218 & lines 299-301).

We also would like to point out that the change in occupancy is not being predicted by early occupancy. The change in occupancy corresponds to the residuals of a regression of the logit of late occupancy vs the logit of early occupancy. Consequently, if one then relates the change in occupancy to early occupancy, there is no significant relationship. This is mentioned in line 146, lines 444-446 & lines 450-452.

To assess the variance explained by different models, we ran four models: Model I, which includes only early occupancy as a predictor of naturalization success; Model II, which incorporates both early occupancy and occupancy change; Model III, which includes early occupancy and woodiness; and Model IV, which combines all three predictors—early occupancy, occupancy change, and woodiness. The calculated Pseudo R² values (by Cragg and Uhler), r2CU showed that models incorporating occupancy change (Model II and Model IV) consistently had higher explanatory power compared to their counterparts (Model I and

Model III), highlighting the role of occupancy change in predicting naturalization success. Although the change in explained variance is not huge, we would like to point out that, as suggested by the previous reviewer #1 (and discussed in lines 238-250), the association between occupancy-change and global naturalization success may be an underestimate. We have now added a supplementary table (TableS24) comparing the pseudo R2 values across all models, and we briefly refer to these results in the main text (lines 216 & 237).

My suggestion would be to change 'region' for countries. Region usually indicates ecological, e.g., climatic, differences, but these data sets reflect political units. Sorting the data into actual ecological regions might have shown more interesting/informative results.

RESPONSE: We thank the reviewer for this suggestion, however we prefer to use 'regions' instead of 'countries' because three of the native regions in our study are only part of countries. Referring, for example, to Thiérache as 'France' would be misleading. However, if the editor prefers us to use 'countries' or another term, we are willing to change it. We now added "(countries or parts thereof) at first mentioning of 'European regions' in the Abstract (line 59) and in the Methods (line384).

It is not clear if changes in occupancy overtime were standardized by the period of time between census, i.e., more change expected if longer time between censuses.

RESPONSE: The changes in occupancy over time are standardized within each of the regions, as the period is the same for all species within a region. We added a note to clarify this at lines 495-496. However, the changes in occupancy over time are not standardized across regions. As we only compared changes in occupancy within regions, a standardization across regions would not change the conclusions.

I can't tell the difference between Table S3 and the ones that follow for each country, information is the same.

RESPONSE: Table S3 (Table S1 in the revised version) provides an overview of the estimates and significances for the analyses that included woodiness and is directly comparable to Table 2 in the main text. Tables S2-S11 (in the revised version) provides the detailed results for each region separately, and we now explain this in the caption of Table S1.

Reviewer #4 (Remarks on code availability):

The analyses are done using standard techniques and standard software packages.